# PROPORTIONAL MULTICALIBRATION

## ABSTRACT

Multicalibration is a desirable fairness criteria that constrains calibration error among flexibly-defined groups in the data while maintaining overall calibration. However, when outcome probabilities are correlated with group membership, multicalibrated models can exhibit a higher percent calibration error among groups with lower base rates than groups with higher base rates. As a result, it remains possible for a decision-maker to learn to trust or distrust model predictions for specific groups. To alleviate this, we propose *proportional multicalibration*, a criteria that constrains the percent calibration error among groups and within prediction bins. We prove that satisfying proportional multicalibration bounds a model's multicalibration as well its *differential calibration*, a stronger fairness criteria inspired by the fairness notion of sufficiency. We provide an efficient algorithm for post-processing risk prediction models for proportional multicalibration and evaluate it empirically. We conduct simulation studies and investigate a real-world application of PMC-postprocessing to prediction of emergency department patient admissions. We observe that proportional multicalibration is a promising criteria for controlling simultenous measures of calibration fairness of a model over intersectional groups with virtually no cost in terms of classification performance.

## 1 INTRODUCTION

Today, machine learning (ML) models have an impact on outcome disparities across sectors (health, lending, criminal justice) due to their wide-spread use in decision-making. When applied in clinical decision-making, ML models help care providers decide whom to prioritize to receive finite and time-sensitive resources among a population of potentially very ill patients. These resources include hospital beds (Barak-Corren et al., 2021a; Dinh & Berendsen Russell, 2021), organ transplants (Schnellinger et al., 2021), specialty treatment programs (Henry et al., 2015; Obermeyer et al., 2019), and, recently, ventilator and other breathing support tools to manage the COVID-19 pandemic (Riviello et al., 2022).

In scenarios like these, decision makers typically rely on risk prediction models to be *calibrated*. Calibration measures the extent to which a model's risk scores, $R$, match the observed probability of the event, $y$. Perfect calibration implies that $P(y|R = r) = r$, for all values of $r$. Calibration allows the risk scores to be used to rank patients in order of priority and informs care providers about the urgency of treatment. However, models that are not equally calibrated among subgroups defined by different sensitive attributes (race, ethnicity, gender, income, etc.) may lead to systematic denial of resources to marginalized groups (e.g. (Obermeyer et al., 2019; Ashana et al., 2021; Roberts, 2011; Zelnick et al., 2021; Ku et al., 2021)). Just this scenario was observed by Obermeyer et al. (2019) analyzed a large health system algorithm used to enroll high-risk patients into care management programs and showed that, at a given risk score, Black patients exhibited significantly poorer health than white patients.

To address equity in calibration, Hebert-Johnson et al. (2018) proposed a fairness measure called *multicalibration* (MC), which asks that calibration be satisifed simultaneously over many flexibly-defined subgroups. Remarkably, MC can be satisfied efficiently by post-processing risk scores without negatively impacting the generalization error of a model, unlike other fairness concepts like demographic parity (Foulds & Pan, 2020) and equalized odds (Hardt et al., 2016). This has motivated the use of MC in practical settings (e.g. Barda et al. (2021)) and has spurred several extensions (Kim et al., 2019; Jung et al., 2021; Gupta et al., 2021; Gopalan et al., 2022). If we bin our risk predictions,

the MC criteria specifies that, for every group within each bin, the absolute difference between the mean observed outcome and the mean of the predictions should be small.

As Barocas et al. (2019) note, equity in calibration embeds the fairness notion called *sufficiency*, which states: for a given risk prediction, the expected outcome should be independent of group membership. Starting from this notion, we can assess the conditions under which MC satisfies sufficiency. In this work, we derive a fairness criteria directly from sufficiency dubbed *differential calibration* for its relation to differential fairness (Foulds et al., 2019b). We show that satisfying differential calibration can ensure that a model is equally "trustworthy" among groups in the data. By equally "trustworthy", we mean that a decision maker cannot reasonably come to distrust the model's risk predictions for specific groups, which may help prevent differences in decision-making between demographic groups, given the same risk prediction.

By relating sufficiency to MC, we describe a shortcoming of MC that can occur when the outcome probabilities are strongly tied to group membership. Under this condition, the amount of calibration error *relative to the expected outcome* can be unequal between groups. This inequality hampers the ability of MC to (approximately) guarantee sufficiency, and thus guarantee equity in trustworthiness for the decision maker.

We propose a simple variant of MC called *proportional multicalibration* (PMC) that ensures that the proportion of calibration error within each bin and group is small. We prove that PMC bounds both multicalibration and differential calibration. We show that PMC can be satisfied with an efficinet post-processing method, similarly to MC.

## 1.1 OUR CONTRIBUTIONS

In this manuscript, we formally analyze the connection of MC to the fairness notion of sufficiency. To do so, we introduce differential calibration (DC), a sufficiency measure that constrains ratios of population risk between pairs of groups within prediction bins. We describe how DC, like sufficiency, provides a sense of equal trustworthiness from the point of view of the decision maker. With this definition, we prove the following. First, models that are $\alpha$-multicalibrated satisfy $(\ln \frac{r_{min}+\alpha}{r_{min}-\alpha})$-DC, where $r_{min}$ is the minimum expected risk prediction among categories defined by subgroups and prediction intervals. We illustrate the meaning of this bound, which is that the proportion of calibration error in multicalibrated models may scale inversely with the outcome probability.

Based on these observations, we propose a variation of MC, PMC, that controls the percentage error by group and risk strata (Definition 5). We show that models satisfying $\alpha$-PMC are $(\frac{\alpha}{1-\alpha})$-multicalibrated and $(\ln \frac{1+\alpha}{1-\alpha})$-differentially calibrated. Proportionally multicalibrated models thereby obtain robust fairness guarantees that are less dependent on population risk categories. Furthermore, we define an efficient algorithm for learning predictors satisfying $\alpha$-PMC.

Finally, we investigate the application of these methods to predicting patient admissions in the emergency department, a real-world resource allocation task, and show that post-processing for PMC results in models that are accurate, multicalibrated, and differentially calibrated.

## 2 RECONCILING MULTICALIBRATION AND SUFFICIENCY

### 2.1 PRELIMINARIES

We consider the task of training a risk prediction model for a population of individuals with outcomes, $y \in \{0, 1\}$, and features, $x \in \mathcal{X}$. Let $D$ be the joint distribution from which individual samples $(y, x)$ are drawn. We assume the outcomes $y$ are random samples from underlying independent Bernoulli distributions, denoted as $p^*(x) \in [0, 1]$. Individuals can be further grouped into *collections of subsets*, $\mathcal{C} \subseteq 2^{\mathcal{X}}$, such that $S \in \mathcal{C}$ is the subset of individuals belonging to $S$, and $x \in S$ indicates that individual $x$ belongs to group $S$.

We denote our risk prediction model as $R(x) : \mathcal{X} \to [0, 1]$. In order to consider calibration in practice, the risk predictions are typically discretized and considered within intervals. The coarseness of this interval is parameterized by a partitioning parameter, $\lambda \in (0, 1)$. The $\lambda$-*discretization* of $[0, 1]$

is denoted by a set of intervals, $\Lambda_\lambda = \left\{ \{I_j\}_{j=0}^{1/\lambda - 1} \right\}$, where $I_j = [j\lambda, (j+1)\lambda)$. For brevity, some proofs in the following sections are given in Appendix A.2.4.

## 2.2 MULTICALIBRATION

MC (Hébert-Johnson et al., 2018) guarantees that the calibration error for any group from a collection of subsets, $\mathcal{C}$ will not exceed a user-defined threshold, over the range of risk scores.

**Definition 1** ($\alpha$-calibration (Hébert-Johnson et al., 2018)). *Let $S \subseteq \mathcal{X}$. For $\alpha \in [0, 1]$, $R$ is $\alpha$-calibrated with respect to $S$ if there exists some $S' \subseteq S$ with $|S'| \geq (1 - \alpha)|S|$ such that for all $r \in [0, 1]$,*

$$\left| \mathbb{E}_D[y|R = r, x \in S'] - r \right| \leq \alpha.$$

**Definition 2** ($\alpha$-MC (Hébert-Johnson et al., 2018)). *Let $\mathcal{C} \subseteq 2^{\mathcal{X}}$ be a collection of subsets of $\mathcal{X}$, $\alpha \in [0, 1]$. A predictor $R$ is $\alpha$-multicalibrated on $\mathcal{C}$ if for all $S \in \mathcal{C}$, $R$ is $\alpha$-calibrated with respect to $S$.*

We note that, according to Definition 1, a model need only be calibrated over a sufficiently large subset of each group ($S'$) in order to satisfy the definition. This relaxation is used to maintain a satisfactory definition of MC when working with discretized predictions. For simplicity, we conduct most of our analysis using the continuous versions of fairenss definitions like Definition 2 (see Appendix A.2.1 for an extended discussion).

MC is one of few approaches to achieving fairness that does not require a significant trade-off to be made between a model's generalization error and the improvement in fairness it provides (Hébert-Johnson et al., 2018). As Hébert-Johnson et al. (2018) show, this is because achieving multicalibration is not at odds with achieving accuracy in expectation for the population as a whole. This separates calibration fairness from other fairness constraints like demographic parity and equalized odds (Hardt et al., 2016), both of which may denigrate the performance of the model on specific groups (Chouldechova, 2017; Pleiss et al., 2017). In clinical settings, such trade-offs may be difficult or impossible to justify. In addition to its alignment with accuracy in expectation, Hébert-Johnson et al. (2018) propose an efficient post-processing algorithm for MC similar on boosting. We discuss additional extensions to MC in Appendix A.1.

## 2.3 SUFFICIENCY AND DIFFERENTIAL CALIBRATION

MC provides a sense of fairness by approximating *calibration by group*, which is perfectly satisfied when $P_D(y|R = r, x \in S) = r$ for all $S \in C$ and $r \in [0, 1]$. Calibration by group is closely related to the *sufficiency* fairness criterion (Barocas et al., 2019). Sufficiency is the condition where the outcome probability is independent from $\mathcal{C}$ conditioned on the risk score. In the binary group setting ($\mathcal{C} = \{S_i, S_j\}$), sufficiency can be expressed as $P_D(y|R, x \in S_i) = P_D(y|R, x \in S_j)$, or

$$\frac{P_D(y|R, x \in S_i)}{P_D(y|R, x \in S_j)} = 1. \tag{1}$$

Unlike calibration by group, sufficiency does not stipulate that the risk scores be calibrated, yet from a fairness perspective, sufficiency and calibration-by-group are equivalent (Barocas et al., 2019). Consider that one can easily transform a model satisfying sufficiency into one that is calibrated-by-group with a single function $f(R) \to [0, 1]$, for example with Platt scaling (Barocas et al., 2019). In both cases, the sense of *fairness* stems from the desire for $R$ to capture everything about group membership that is relevant to predicting $y$.

Under sufficiency, the risk score is equally informative of the outcome, regardless of group membership. In this sense, a model satisfying sufficiency provides *equally trustworthy* risk predictions to a decision maker, regardless of the groups to which an individual belongs. Below, we define an approximate measure of sufficiency that constrains pairwise differentials between groups, and accomodates binned predictions:

**Definition 3** ($\varepsilon$-Differential Calibration). *Let $\mathcal{C} \subseteq 2^{\mathcal{X}}$ be a collection of subsets of $\mathcal{X}$. A model $R(x)$ is $\varepsilon$-differentially calibrated with respect to $\mathcal{C}$ if, for all pairs $(S_i, S_j) \in \mathcal{C} \times \mathcal{C}$ for which $P_D(S_i), P_D(S_j) > 0$, for any $r \in [0, 1]$,*

$$e^{-\varepsilon} \leq \frac{\mathbb{E}_D[y|R = r, x \in S_i]}{\mathbb{E}_D[y|R = r, x \in S_j]} \leq e^{\varepsilon} \tag{2}$$

By inspection we see that $\varepsilon$ in $\varepsilon$-DC measures the extent to which $R$ satisifies sufficiency. That is, when $P(y|R = r, x \in S_i) \approx P(y|R = r, x \in S_j)$ for all pairs $(S_i, S_j)$, $\varepsilon \approx 0$. $\varepsilon$-DC says that, within any bin of risk scores, the outcome $y$ is at most $e^{\varepsilon}$ times more likely among one group than another, and a minimum of $e^{-\varepsilon}$ less likely. Definition 13 fits into the general definition of a *differential fairness* measure proposed by Foulds et al. (2019a), although previously it was used to define demographic parity criteria (Foulds & Pan, 2020). We describe the relation in more detail in Appendix A.1.1, including Eq. (2)'s connection to differential privacy Dwork & Lei (2009) and pufferfish privacy Kifer & Machanavajjhala (2014).

### 2.4 THE DIFFERENTIAL CALIBRATION OF MULTICALIBRATED MODELS IS LIMITED BY LOW-RISK GROUPS

At a basic level, the form of MC and sufficiency differ: MC constrainins absolute differences between groups across prediction bins, whereas sufficiency constrains pairwise differentials between groups. To reconcile MC and DC/sufficiency more formally, we pose the following question: if a model satisfies $\alpha$-MC, what, if anything does this imply about the $\varepsilon$-DC of the model? (In Appendix A.3, Theorem 4, we answer the inverse question). We now show that multicalibrated models have a bounded DC, but that this bound is limited by small values of $R$.

**Theorem 1.** *Let $R(x)$ be a model satisfying $\alpha$-MC on a collection of subsets $\mathcal{C} \in 2^{\mathcal{X}}$. Let $r_{min} = \min_{S \in \mathcal{C}} \mathbb{E}_D[R|R = r, x \in S]$ be the minimum expected risk prediction among $S \in \mathcal{C}$ and $r \in [0, 1]$. Then R(x) is $\left(\ln \frac{r_{min}+\alpha}{r_{min}-\alpha}\right)$-differentially calibrated.*

*Proof.* Let $r = \mathbb{E}_D[R|R = r, x \in S]$ and $p^* = \mathbb{E}_D[y|R = r, x \in S]$. $\alpha$-MC guarantees that $r - \alpha \leq p^* \leq r + \alpha$ for all groups $S \in \mathcal{C}$ and predictions $r \in [0, 1]$. Plugging these lower and upper bounds into Eq. (2), we observe that the lower bound on $\varepsilon$-DC for $R(x)$ is given by $\frac{r+\alpha}{r-\alpha} \leq e^{\varepsilon}$. The maximum of the left-hand side for a fixed $\alpha$ occurs at the smallest value of $r$; therefore $R(x)$ satisfies $\ln \frac{r_{min}+\alpha}{r_{min}-\alpha} \leq \varepsilon$. By switching the numeratator and denominator we obtain the minimum differential and the left-hand side constraint from Definition 13, i.e. $e^{-\varepsilon} \leq \frac{r_{min}-\alpha}{r_{min}+\alpha}$. Thus $R(x)$ is $\left(\ln \frac{r_{min}+\alpha}{r_{min}-\alpha}\right)$-differentially calibrated. $\square$ $\square$

Theorem 1 illustrates the important point that, *in terms of percentage error*, $MC$ does not provide equal protection to groups with different risk profiles. Imagine a model satisfying (0.05)-MC for groups $S \in \mathcal{C}$. Consider individuals receiving model predictions $R(x) = 0.9$. MC guarantees that, for any category $\{x : x \in S, R(x) = 0.9\}$, the expected outcome probability is at least $0.9 - \alpha = 0.85$ and at most $0.9 + \alpha = 0.95$. This bounds the percent error among groups with this prediction to about 6%. In contrast, consider individuals for whom $R(x) = 0.3$; each group may have a true outcome probability as low as 0.25, which is an error of 20% - about 3.4x higher than the percent error in the higher-risk group.

## 3 PROPORTIONAL MULTICALIBRATION

We are motivated to define a measure that is efficiently learnable like MC (Definition 2) but better aligned with the multiplicative interpretation of sufficiency, like DC (Definition 13). To do so, we define PMC, a variant of MC that constrains the proportional calibration error of a model among subgroups and risk strata. In this section, we show that bounding a model's PMC is enough to meaningfully bound DC and MC. Furthermore, we provide an efficient algorithm for satisfying PMC based on a simple extension of MC/Multiaccuracy boosting (Kim et al., 2019). We begin by

defining proportional calibration, which expresses calibration error as a percentage of the outcome probability among a group.

**Definition 4** ($\alpha$-Proportional Calibration). *Let $S \subseteq \mathcal{X}$. For $\alpha > 0$, $R$ is $\alpha$-proportionally calibrated with respect to $S$ if there exists some $S' \subseteq S$ with $|S'| \geq (1 - \alpha) |S|$ such that for all $r \in [0, 1]$,*

$$\left| \mathbb{E}_D[y|R = r, x \in S'] - r \right| \leq \alpha \; \mathbb{E}_D[y|R = r, x \in S'].$$

Proportional multicalibration is then defined by requiring Definition 4 be satisified among a collection of groups:

**Definition 5** ($\alpha$-Proportional Multicalibration). *Let $\mathcal{C} \subseteq 2^{\mathcal{X}}$ be a collection of subsets of $\mathcal{X}$, $\alpha > 0$. A predictor $R$ is $\alpha$-proportionally multicalibrated on $\mathcal{C}$ if for all $S \in \mathcal{C}$, $R$ is $\alpha$-proportionally calibrated with respect to $S$.*

We also define a discretized version of PMC that is useful for implementing the measure in Algorithm 1 and measuring PMC in our experiments.

**Definition 6** (($\alpha, \lambda$)-Proportional Multicalibration). *A model $R(x)$ is ($\alpha, \lambda$)-proportionally multicalibrated with respect to a collection of subsets $\mathcal{C}$ if, for all $S \in \mathcal{C}$ and $I \in \Lambda_\lambda$ satisfying $P_D(R(x) \in I | x \in S) \geq \alpha\lambda$,*

$$\frac{|\mathbb{E}_D[y|R \in I, x \in S] - \mathbb{E}_D[R|R \in I, x \in S]|}{\mathbb{E}_D[y|R \in I, x \in S]} \leq \alpha. \tag{3}$$

Note that, in practice, we must ensure $\mathbb{E}_D[y|R \in I, x \in S] \geq 0$ for Definition 6 to be satisfied; we later introduce a lower bound, $\rho$, to prevent the outcome probability from being too small. In Appendix A.2.1, we show ($\alpha, \lambda$)-PMC meaningfully bounds $\alpha$-PMC under different discretizations, such that we can minimize ($\alpha, \lambda$)-PMC to achieve low $\alpha$-PMC.

**Comparison to Differential Calibration**    Rather than constraining the differentials of prediction- and group- specific outcomes among all pairs of subgroups in $\mathcal{C} \times \mathcal{C}$ as in DC (Definition 13), PMC constrains the relative error of each group in $\mathcal{C}$. In practical terms, this makes it more efficient to calculate PMC by a factor of $O(|\mathcal{C}|)$ steps compared to DC. In addition, PMC does not require additional assumptions about the overall calibration of a model in order to imply guarantees of MC, since PMC directly constrains calibration rather than constraining sufficiency alone.

**Theorem 2.** *Let $R(x)$ be a model satisfying ($\alpha$)-PMC on a collection $\mathcal{C}$. Then $R(x)$ is ($\ln \frac{1+\alpha}{1-\alpha}$)-differentially calibrated.*

*Proof.* Let $r = \mathbb{E}_D[R|R = r, x \in S]$ and $p^* = \mathbb{E}_D[y|R = r, x \in S]$. If $R(x)$ satisfies $\alpha$-PMC (Definition 5), then $r/(1 + \alpha) \leq p^* \leq r/(1 - \alpha)$. Solving for the upper bound on $\varepsilon$-DC, we immediately have $\varepsilon \geq \ln \frac{r(1+\alpha)}{r(1-\alpha)} \geq \ln \frac{1+\alpha}{1-\alpha}$. $\qquad\square\qquad\qquad\square$

Theorem 2 demonstrates that $\alpha$-proportionally multicalibrated models satisfy a straightforward notion of differential fairness that depends monotonically only on $\alpha$. The relationship between PMC and DC is contrasted with the relationship of MC and DC in Fig. 3, left panel. The figure illustrates how MC's sensitivity to small risk categories limits its DC.

**Comparison to Multicalibration**    Rather than constraining the absolute difference between risk predictions and the outcome as in MC, PMC requires that the calibration error be a small fraction of the expected risk in each category $(S, I)$. In this sense, it provides a stronger protection than MC by requiring calibration error to be a small fraction regardless of the risk group. In many contexts, we would argue that this is also more aligned with the notion of fairness in risk prediction contexts. Under MC, the underlying probability of an outcome within a group affects the fairness protection that is received (i.e., the percentage error that Definition 2 allows). Because underlying probabilities of many clinically relevant outcomes vary significantly among subpopulations, multicalibrated models may systematically permit higher percentage error to specific risk groups. The difference in relative calibration error among populations with different risk profiles also translates in weaker sufficiency guarantees, as demonstrated in Theorem 1. In contrast, PMC provides a fairness guarantee that is less dependent on subpopulation risks. In the following theorem, we show that MC is also constrained when a model satisfies PMC.

---

**Algorithm 1** Proportional Multicalibration Post-processing

---

**Require:** Predictor $R(x)$
1:   $\mathcal{C} \in 2^{\mathcal{X}}$ such that for all $S \in \mathcal{C}, P_D(S) \geq \gamma$
2:   $\alpha, \lambda, \gamma, \rho > 0$
3:   $\mathcal{D} = \{(y,x)_i\}_{i=0}^N \sim D$

4:   **function** PMC($R, \mathcal{C}, \mathcal{D}, \alpha, \lambda, \gamma, \rho$)
5:     **repeat**
6:       $\{(y,x)\} \leftarrow$ sample $\mathcal{D}$
7:       **for** $S \in \mathcal{C}, I \in \Lambda_\lambda$ such that $P_D(R \in I, x \in S) \geq \alpha\lambda\gamma$ **do**
8:         $S_r \leftarrow S \cap \{x : R(x) \in I\}$
9:         $\bar{r} \leftarrow \frac{1}{|S_r|} \sum_{x \in S_r} R(x)$                                   ▷ average group prediction
10:         $\bar{y} \leftarrow \frac{1}{|S_r|} \sum_{x \in S_r} y(x)$                               ▷ average subgroup risk
11:         **if** $\bar{y} \leq \rho$ **then**
12:           continue
13:         $\Delta r \leftarrow \bar{y} - \bar{r}$
14:         **if** $|\Delta r| \geq \alpha\bar{y}$ **then**
15:           $R(x) \leftarrow R(x) + \Delta r$ for all $x \in S_r$
16:           $R(x) \leftarrow$ squash($R(x), [0,1]$)                      ▷ squash updates to $[0,1]$
17:       **if** No Updates to R(x) **then**
18:         break
19:     **return** $R$

---

**Theorem 3.** *Let $R(x)$ be a model satisfying $\alpha$-PMC on a collection $\mathcal{C}$. Then $R(x)$ is $(\frac{\alpha}{1-\alpha})$-multicalibrated on $\mathcal{C}$.*

The proof of Theorem 3 is given in Appendix A.2.4. This theorem implies that a proportionally calibrated model with sufficiently low $\alpha$ will satisfy a similarly low value of MC, whereas a model with 0.5-PMC (i.e., 50% calibration error) or higher will not have a meaningful MC guarantee. We further discuss and illustrate the bounds given by Theorems 1 to 3 and 4 in Appendix A.2.

### 3.1 LEARNING PROPORTIONALLY MULTICALIBRATED PREDICTORS

So far we have demonstrated that models satisfying PMC exhibit desirable guarantees relative to two previously defined measures of fair calibration, but have not considered whether PMC is possible to learn. Here, we answer in the affirmative by proposing Algorithm 1 to satisfy PMC . Algorithm 1 is a direct extension of MCBoost(Pfisterer et al., 2021) , which works by checking for calibration errors among groups and prediction intervals that violate the user threshold and adjusting these predictions towards the target. PMC post-processing differs in two main ways: first, it updates whenever calibration error is not within $\alpha\bar{y}$ for all categories, as opposed to simply within $\alpha$. Second, it ignores updates for categories with low outcome probability (less than $\rho$). Next, we prove that PMC post-processing learns an $(\alpha,\lambda)$-PMC model in a polynomial number of steps.

**Proposition 1.** *Define $\alpha, \lambda, \gamma, \rho > 0$. Let $\mathcal{C} \subseteq 2^{\mathcal{X}}$ be a collection of subsets of $\mathcal{X}$ such that, for all $S \in \mathcal{C}$, $P_D(S) > \gamma$. Let $R(x)$ be a risk prediction model to be post-processed. For all $(S,I) \in \mathcal{C} \times \Lambda_\lambda$, let $E[y|R \in I, x \in S] > \rho$. There exists an algorithm that satisfies $(\alpha,\lambda)$-PMC with respect to $\mathcal{C}$ in $O(\frac{|C|}{\alpha^3\lambda^2\rho^2\gamma})$ steps.*

We analyze Algorithm 1 and show it satisfies Proposition 1 in Appendix A.2.4. This more stringent threshold requires an additional $O(\frac{1}{\rho^2})$ steps, where $\rho > 0$ is a lower bound on the expected outcome within a category $(S,I)$.

## 4 EXPERIMENTS

In our first set of experiments (Section 4), we study MC and PMC in simulated population data to understand and validate the analysis in previous sections. In the second section, we compare the performance of varied model treatments on a real world hospital admission task, using an implementation of Algorithm 1. We make use of empirical versions of our fairness definitions which we refer to as *MC loss*, *PMC loss*, and *DC loss*. In short, these measures calculate the maximum (proportional) calibration error or pairwise calibration differential among subgroups and risk categories in the data sample. Due to space constraints the formal definitions are given in Appendix A.2.2 (Definitions 14 to 16).

**Simulation study** We simulate data from $\alpha$-multicalibrated models. For simplicity, we specify a data structure with a one-to-one correspondence between subset and model estimated risk, such that for all $x$ in $S$, $R(x) = R(x|x \in S) = R(S)$. Therefore all information for predicting the outcome based on the features in $x$ is contained in the attributes $\mathcal{A}$ that define subgroup $S$. Outcome probability is specified as $p_i^* = P_D(y|x \in S_i) = 0.2 + 0.01(i - 1)$ and $i = 1, \cdots, N_s$, where $N_s$ is the number of subsets $S$, defined by $\mathcal{A}$ and indexed by $i$ with increasing $p^*$. For each group, $R_i = R(S_i) = R(x|x \in S_i) = p_i^* - \Delta_i$. We randomly select $\Delta_i$ for one group to be $\pm\alpha$ and for the remaining groups, $\Delta_i = \pm\delta$, where $\delta \sim \text{Uniform}(\min = 0, \max = \alpha)$. In all cases, the sign of $\Delta_i$ is determined by a random draw from a Bernoulli distribution. For these simulations we set $N_S = 61$ and $\alpha = 0.1$, such that $p_i^* \in [0.2, 0.8]$ and $R_i \in [0.1, 0.9]$. We generate $N_{sim} = 1000$ simulated datasets, with $n = 1000$ observations per group, and for each $S_i$, we calculate the ratio of the absolute mean error to $p_i^*$, i.e. the PMC loss function for this data generating mechanism.

We also simulate three specific scenarios where: 1) $|\Delta_i|$ is equivalent for all groups (Fixed); 2) $|\Delta_i|$ increases with increasing $p_i^*$; and 3) $|\Delta_i|$ decreases with increasing $p_i^*$, with $\alpha = 0.1$ in each case. These scenarios compare when $\alpha$ is determined by all groups, the group with the lowest outcome probability, and the group with the highest outcome probability, respectively.

**Hospital admission** Next, we test PMC alongside other methods in application to prediction of inpatient hospital admission for patients visiting the emergency department (ED). The burden of overcrowding and long wait times in EDs is significantly higher among non-white, non-Hispanic patients and socio-economically marginalized patients (James et al., 2005; McDonald et al., 2020). Recent work has demonstrated risk prediction models that can expedite patient visits by predicting patient admission at an early stage of a visit with a high degree of certainty (AUC $\geq 0.9$ across three large care centers) (Barak-Corren et al., 2017b;a; 2021b;a). Our goal is to ensure no group of patients will be over- or under-prioritized over another by these models, which could exacerbate the treatment and outcome disparities that currently exist.

We construct a prediction task similar to previous studies but using a new data resource: the MIMIC-IV-ED repository (Johnson et al., 2021). The overall intersectional demographic statistics for these data are given in Table 1. In Table 1 we observe stark differences in admission rates by demographic group and gender, suggesting that the use of a proportional measure of calibration could be appropriate for this task. We trained and evaluated logistic regression (LR) and random forest (RF) models of patient admission, with and without post-processing for MC (Pfisterer et al., 2021) or PMC. For each of the parameter settings given in Table 2, we conducted 100 repeat experiments with different shuffles of the data. Comparisons are reported on a test set of 20% of the data for each trial. Additional experiment details are available in Appendix A.6 and code for the experiments is available here: `https://github.com/by1tTZ4IsQkAO80F/pmc`.

## 5 RESULTS

Fig. 1 shows the PMC loss of $\alpha$-multicalibrated models under the scenarios described in Section 4. Proportional $\alpha$-MC constrains the ratio of the absolute mean error (AME) to the outcome prevalence, for groups defined by a risk interval ($R(x) \in I$) and subset within a collection of subsets ($x \in S, S \in \mathcal{C}$). Without the proportionality factor $|\mathbb{E}_D[y|R = r, x \in S]|^{-1}$, $\alpha$-multicalibrated models allow a dependence between the group prevalence and the error or privacy loss permitted that is unfair for groups with lower outcome prevalence.

Results on the hospital admission prediction task are summarized in Fig. 2 and Tables 3 and 4. PMC post-processing has a negligible effect on predictive performance ($<0.1\%$ $\Delta$ AUROC, LR and RF) while reducing DC loss by 27% for LR and RF models, and reducing PMC loss by 40% and 79%, respectively. In the case of RF models, PMC post-processing reduces MC loss by 23%, a significantly larger improvement than MC post-processing itself (19%, $p$=9e-26).

**Senstivity Analysis** We note PMC postprocessing has a lower tolerance for error than MC post-processing for a given value of $\alpha$, since $\bar{y} \leq 1$. Thus a natural question is whether MC can match the performance of PMC simply by specifying a smaller $\alpha$. Further, if $\alpha$ can be made small enough, the calibration error $|\mathbb{E}_D[R|R \in I, x \in S] - \mathbb{E}_D[y|R \in I, x \in S]|$ on all categories will be small compared to the outcome prevalence, $\mathbb{E}_D[y|R \in I, x \in S]$, meaning MC postprocessing could

Table 1: Admission prevalence (Admissions/Total (%)) among patients in the MIMIC-IV-ED data repository, stratified by the intersection of ethnoracial group and gender.

| Gender
Ethnoracial Group | F | M | Overall |
|---|---|---|---|
| American Indian/Alaska Native | 70/257 (27%) | 82/170 (48%) | 152/427 (36%) |
| Asian | 1043/3595 (29%) | 1032/2384 (43%) | 2075/5979 (35%) |
| Black/African American | 3124/27486 (11%) | 2603/14458 (18%) | 5727/41944 (14%) |
| Hispanic/Latino | 1063/10262 (10%) | 1168/5795 (20%) | 2231/16057 (14%) |
| Other | 1232/5163 (24%) | 1479/3849 (38%) | 2711/9012 (30%) |
| Unknown/Unable to Obtain | 1521/2156 (71%) | 2074/2377 (87%) | 3595/4533 (79%) |
| White | 18147/50174 (36%) | 18951/45435 (42%) | 37098/95609 (39%) |
| Overall | 26200/99093 (26%) | 27389/74468 (37%) | 53589/173561 (31%) |

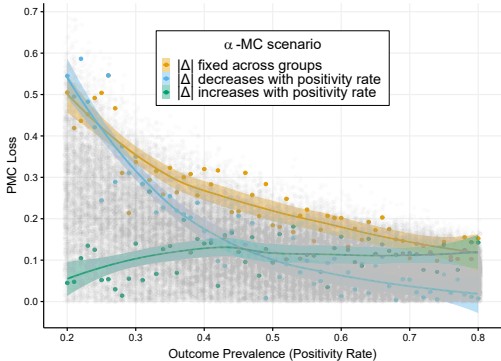

Figure 1: The relationship between MC, PMC, and outcome prevalence as illustrated via a simulation study in which the rates of the outcome are associated with group membership. Gray points denote the PMC loss of a (0.1,0.1)-MC model on 1000 simulated datasets, and colored lines denote three specific scenarios in which each group's calibration error ($|\Delta|$) follows specific rules. PMC loss is higher among groups with lower positivity rates in most scenarios unless the groupwise calibration error increases with positivity rate.

satisfy the desired proportional guarantees of PMC. However, to achieve this performance by MC-postprocessing may require a large number of unnecessary updates for high risk intervals, since the DC and PMC of multicalibrated models are limited by low-risk groups (Theorem 1). Furthermore, the number of steps in MC-postprocessing (and PMC-postprocessing) scales as an inverse high-order polynomial of $\alpha$ (cf. Thm. 2 (Hebert-Johnson et al., 2018)).

We shed light on this question in three ways. First, we quantify how often the use of each post-processing algorithm gives the best loss for each metric and trial in Table 3. To account for this, we quantified the number of trials for which a given method produced the best model according to a given metric, over all parameter configurations in Table 2. PMC post-processing (Algorithm 1) achieves the best fairness the highest percent of the time, according to DC loss (63%), MC loss (70%), and PMC loss (72%), while MC-postprocessed models achieve the best AUROC in 88% of cases. This provides strong evidence that, over a large range of $\alpha$ values, PMC post-processing is beneficial compared to MC-postprocessing.

Next, we empirically compare MC- and PMC-postprocessing by the number of steps required for each to reach their best performance in Fig. 8 and Table 4. We assess how many steps/updates MC and PMC take, and their empirical running time, in Table 4. We observe that PMC typically requires a larger number of updates to achieve its best performance on MC loss (about 2x wall clock time and number of updates), In contrast MC-postprocessing requires an average of 5x more updates to achieve its best performance on PMC loss, due to its dependence on very small values of $\alpha$.

Finally, in Appendix A.6, we look at detailed performance comparisons of MC and PMC postprocessing over values of $\alpha$ and group definitions in Figs. 5 to 7. We observe that, while low values of $\alpha$ for MC postprocessing improve its PMC loss performance, PMC postprocessing is always able to outperform it for some value of $\alpha$, and do so in fewer steps.

## 6 DISCUSSION AND CONCLUSION

In this paper we have analyzed multicalibration through the lens of sufficiency and differential calibration to reveal the sensitivity of this metric to correlations between outcome rates and group membership. We have proposed a measure, PMC, that alleviates this sensitivity and attempts to

Table 2: Parameters for the hospital admission prediction experiment.

| Parameter | Values |
|---|---|
| $\alpha$ | (0.001, 0.01, 0.05, 0.1) |
| $\gamma$ | (0.05, 0.1) |
| $\lambda$ | 0.1 |
| $\rho$ | (0.001, 0.01) |
| Model | LR, RF |
| Groups | [(race/ethnicity, gender), (race/ethnicity, gender, insurance product)] |

Table 3: The number of times each post-processing method achieved the best score among all methods, out of 100 trials.

| postprocessing metric | Base Model | MC | PMC |
|---|---|---|---|
| AUROC | 5 | 88 | 6 |
| DC loss | 0 | 36 | 63 |
| MC loss | 8 | 21 | 70 |
| PMC loss | 0 | 27 | 72 |

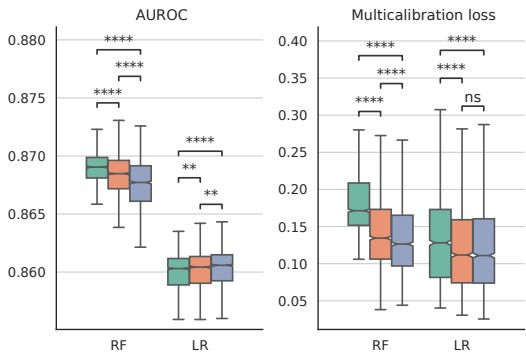 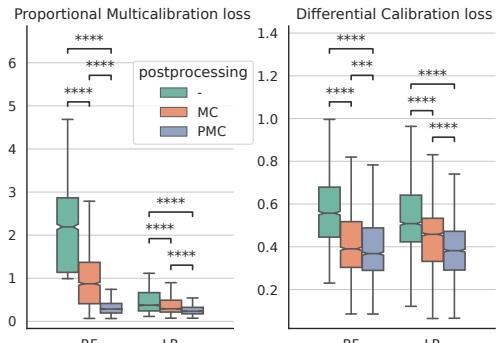

Figure 2: A comparison of LR and RF models, with and without MC and PMC post-processing, on the hospital admission task. From left to right, trained models are compared in terms of test set AUROC, MC loss, PMC loss, and DC loss. Points represent the median performance over 100 shuffled train/test splits with bootstrapped 99% confidence intervals. We test for significant differences between post-processing methods using two-sided Wilcoxon rank-sum tests with Bonferroni correction. ns: $p <= 1$; **: 1e-03 $< p <=$ 1e-02; ***: 1e-04 $< p <=$ 1e-03; ****: $p <=$ 1e-04.

capture the "best of both worlds" of MC and DC. PMC provides equivalent percentage calibration protections to groups regardless of their risk profiles, and in so doing, bounds a model's differential calibration. We provide an efficient algorithm for learning PMC predictors by postprocessing a given risk prediction model. On a real-world and clinically relevant task (admission prediction), we have shown that post-processing LR and RF models with PMC leads to better performance across all three fairness metrics, with little to no impact on predictive performance.

Our preliminary analysis suggests PMC can be a valuable metric for training fair algorithms in resource allocation contexts. Future work could extend this analysis on both the theoretical and practical side. On the theoretical side, the generalization properties of the PMC measure should be established and its sample complexity quantified, as Rose (2018) did with MC. Additional extensions of PMC could establish a bound on the accuracy of PMC-postprocessed models in a similar vein to work by Kim et al. (2019) and Hébert-Johnson et al.. On the empirical side, future works should benchmark PMC on a larger set of real-world problems, and explore use cases in more depth.

Table 4: For MC- and PMC-postprocessing, we compare the median number of updates and median wall clock time (s) taken to train for the hyperparameter settings (e.g. $\alpha$) that achieved the best performance on each metric.

| Metric | ML | Postprocessing | Best Loss | # of Updates | Wall Clock Time (s) |
|---|---|---|---|---|---|
| MC loss | LR | MC | 0.116 | 0 | 32.2 |
| | | PMC | 0.108 | 30 | 58.1 |
| | RF | MC | 0.147 | 82 | 44.3 |
| | | PMC | 0.135 | 172 | 79.6 |
| PMC loss | LR | MC | 0.334 | 376 | 110.6 |
| | | PMC | 0.287 | 52 | 66.7 |
| | RF | MC | 0.356 | 504 | 106.5 |
| | | PMC | 0.325 | 188 | 81.7 |

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

# A    APPENDIX

In this section, we include additional comparisons to related work, additional definitions, proofs to the theorems in the main text, and additional experimental details. The code to reproduce the figures and experiments is available here: https://github.com/by1tTZ4IsQkAO80F/pmc.

## A.1    RELATED WORK

**Definitions of Fairness**    There are myriad ways to measure fairness that are covered in more detail in other works (Barocas et al., 2019; Chouldechova & Roth, 2018; Castelnovo et al., 2021). We briefly review three notions here. The first, *demographic parity*, requires the model's predictions to be independent of patient demographics ($A$). Although a model satisfying demographic parity can be desirable when the outcome should be unrelated to sensitive attributes (Foulds & Pan, 2020), it can be unfair if important risk factors for the outcome are associated with those attributes (Hardt et al., 2016). For example, it may be more fair to admit socially marginalized patients to a hospital at a higher rate if they are assessed less able to manage their care at home. Furthermore, if the underlying rates of illness vary demographically, requiring demographic parity can result in a healthier patients from one group being admitted more often than patients who urgently need care.

When the base rates of admission are expected to differ demographically, we can instead ask that the model's errors be balanced across groups. One such notion is *equalized odds*, which states that for a given $Y$, the model's predictions should be independent of $A$. Satisfying equalized odds is equivalent to having equal FPR and FNR for every group in $A$.

When the model is used for patient risk stratification, as in the target use case in this paper, it is important to consider a model's calibration for each demographic group in the data. Because risk prediction models influence who is prioritized for care, an unfairly calibrated model can systematically under-predict risk for certain demographic groups and result in under-allocation of patient care to those groups. Thus, guaranteeing group-wise calibration via an approach such as multicalibration also guarantees fair patient prioritization for health care provision. In some contexts, risk predictions are not directly interpreted, but only used to *rank* patients, which in some contexts is sufficient for resource allocation. Authors have proposed various ways of measuring the fairness of model rankings, for example by comparing AUROC between groups (Kallus et al., 2020).

**Approaches to Fairness**    Many approaches to achieving fairness guarantees according to demographic parity, equalized odds and its relaxations have been proposed (Dwork et al., 2012; Hardt et al., 2016; Berk et al., 2017; Jiang & Nachum, 2019; Kearns et al., 2018). When choosing an approach, is important to carefully weigh the relative impact of false positives, false negatives, and miscalibration on patient outcomes, which differ by use case. When group base rates differ (i.e., group-specific positivity rates), *equalized odds and calibration by group cannot both be satisfied* (Kleinberg et al., 2016). Instead, one can often equalized multicalibration while satisfying relaxations of equalized odds such as *equalized accuracy*, where $Accuracy = \mu TPR + (1-\mu)(1-FPR)$ for a group with base rate $\mu$. However, to do so requires denigrating the performance of the model on specific groups (Chouldechova, 2017; Pleiss et al., 2017), which is unethical in our context.

As mentioned in the introduction, we are also motivated to utilize approaches to fairness that 1) dovetail well with intersectionality theory, and 2) provide privacy guarantees. Most work in the computer science/ machine learning space does not engage with the broader literature on socio-cultural concepts like intersectionality, which we see as a gap that makes adoption in real-world settings difficult (Hanna et al., 2020). One exception to this statement is differential fairness (Foulds et al., 2019a), a measure designed with intersectionality in mind. In addition to being a definition of fairness that provides equal protection to groups defined by intersections of protected attributes, models satisfying $\varepsilon$-differential fairness also satisfy $\varepsilon$-pufferfish privacy. This privacy guarantee is very desirable in risk prediction contexts, because it limits the extent to which the model reveals sensitive information to a decision maker that has the potential to influence their interpretation of the model's recommendation. However, prior work on differential fairness has been limited to using it to control for demographic parity, which is not an appropriate fairness measure for our use case (Foulds & Pan, 2020).

Multicalibration has inspired several extensions, including relaxations such as multiaccuracy (Kim et al., 2019), low-degree multicalibration (Gopalan et al., 2022), and extensions to conformal prediction and online learning (Jung et al., 2021; Gupta et al., 2021). Noting that multicalibration is a guarantee over mean predictions on a collection of groups $\mathcal{C}$, Jung et al. (2021) propose to extend multicalibration to higher-order moments (e.g., variances), which allows one to estimate a confidence interval for the calibration error for each category. Gupta et al. (2021) extend this idea and generalize it to the online learning context, in which an adversary chooses a sequence of examples for which one wishes to quantify the uncertainty of different statistics of the predictions. Recent work has also utilized higher order moments to "interpolate" between the guarantees provided by multiaccuracy, which only requires accuracy in expectation for groups in $\mathcal{C}$, and multicalibration, which requires accuracy in expectation at each prediction interval (Kim et al., 2019). Like proportional multicalibration (Definition 5), definitions of multicalibration for higher order moments provide additional criteria for quantifying model performance over many groups; in general, however, much of the focus in other work is on statistics for uncertainty estimation. Like these works, one may view our proposal for proportional multicalibration as alternative definition of what it means to be multicalibrated. The key difference is that proportional multicalibration measures the degree to which multicalibration depends on differences in outcome prevalence between groups, and in doing so provides guarantees of pufferfish privacy and differential calibration.

Dwork et al. (2019) study the relation of fair rankings to multicalibration, and, in a similar vein to differential fairness measures, formulate a fairness measure for group rankings using the relations between pairs of groups. However, these definitions are specific to the ranking relation between the groups, whereas differential calibration cares only about the outcome differential (conditioned on model predictions) between pairs of groups.

### A.1.1 DIFFERENTIAL FAIRNESS

DF was explicitly defined to be consistent with the social theoretical framework of *intersectionality*. This framework dates back as early as the social movements of the '60s and '70s (Collins & Bilge, 2020) and was brought into the academic mainstream by pioneering work from legal scholar Kimberlé Crenshaw (Crenshaw, 1989; 1991) and sociologist Patricia Hill Collins (Collins, 1990). Central to intersectionality is that hierarchies of power and oppression are structural elements that are fundamental to our society. Through an intersectional lens, these power structures are viewed as interacting and co-constituted, inextricably related to one another. To capture this viewpoint, DF (Foulds et al., 2019a) constrains the differential of a general data mechanism among all pairs of groups, where groups are explicitly defined as the intersections of protected attributes in $\mathcal{A}$.

**Definition 7** ($\varepsilon$-differential fairness (Foulds et al., 2019a))**.** *Let $\Theta$ denote a set of distributions and let $x \sim \theta$ for $\theta \in \Theta$. A mechanism $M(x)$ is $\varepsilon$-differentially fair with respect to $(\mathcal{C}, \Theta)$ for all $\theta \in \Theta$ with $x \sim \theta$, and $m \in Range(M)$ if, for all $(S_i, S_j) \in \mathcal{C} \times \mathcal{C}$ where $P(S_i|\theta) > 0$, $P(S_j|\theta) > 0$,*

$$e^{-\varepsilon} \leq \frac{P_{M,\theta}(M(x) = m|S_i, \theta)}{P_{M,\theta}(M(x) = m|S_j, \theta)} \leq e^{\varepsilon} \tag{4}$$

**Definition 8** (Pufferfish Privacy)**.** *Let the collection of subsets $\mathcal{C}$ represent sets of secrets. A mechanism $M(x)$ is $\varepsilon$-pufferfish private (Kifer & Machanavajjhala, 2014) with respect to $(\mathcal{C}, \Theta)$ if for all $\theta \in \Theta$ with $x \sim \theta$, for all secret pairs $(S_i, S_j) \in \mathcal{C} \times \mathcal{C}$ and $y \in Range(M)$,*

$$e^{-\varepsilon} \leq \frac{P_{M,\theta}(M(x) = y|S_i, \theta)}{P_{M,\theta}(M(x) = y|S_j, \theta)} \leq e^{\varepsilon}, \tag{5}$$

*when $S_i$ and $S_j$ are such that $P(S_i|\theta) > 0$, $P(S_j|\theta) > 0$.*

**Note on pufferfish and differential privacy** Although Eq. (4) is notable in its similarity to differential privacy (Dwork & Lei, 2009), they differ in important ways. Differential privacy aims to limit the amount of information learned about any one individual in a database by computations performed on the data (e.g. $M(x)$). Pufferfish privacy only limits information learned about the group membership of individuals as defined by $\mathcal{C}$. Kifer & Machanavajjhala (2014) describe in detail the conditions under which these privacy frameworks are equivalent.

**Efficiency Property** Foulds et al. (2019a) also define an interesting property of $\varepsilon$-differential fairness that allows guarantees of higher order (i.e., marginal) groups to be met for free; the property is given in Appendix A.2.2.

**Definition 9** (Efficiency Property (Foulds et al., 2019a)). *Let $M(x)$ be an $\varepsilon$-differentially fair mechanism with respect to $(\mathcal{C}, \Theta)$. Let the collection of subsets $\mathcal{C}$ group individuals according to the Cartesian product of attributes $A \subseteq \mathcal{A}$. Let $\mathcal{G}$ be any collection of subsets that groups individuals by the Cartesian product of attributes in $A'$, where $A' \subset A$ and $A' \neq \emptyset$. Then $M(x)$ is $\varepsilon$-differentially fair in $(\mathcal{G}, \Theta)$.*

The authors call this the "intersectionality property", although its implication is the reverse: if a model satisfies $\varepsilon$-DF for the low level (i.e. intersectional) groups in $\mathcal{C}$, then it satisfies $\varepsilon$-DF for every higher-level (i.e. marginal) group. For example, if a model is $(\varepsilon)$-differentially fair for intersectional groupings of individuals by race and sex, then it is $\varepsilon$-DF for the higher-level race and sex groupings as well. Whereas the number of intersections grows exponentially as additional attributes are protected (Kearns et al., 2018), the number of total possible subgroupings grows at a larger combinatorial rate: for $p$ protected attributes, we have $\sum_{k=1}^{p} \binom{p}{k} m_a^k$ groups, where $m_a$ is the number of levels of attribute $a$.

**Limitations** To date, analysis of DF for predictive modeling has been limited to defining $R(x)$ as the mechanism, which is akin to asking for *demographic parity*. Under demographic parity, one requires that model predictions be independent from group membership entirely, and this limits the utility of it as a fairness notion. Although a model satisfying demographic parity can be desirable when the outcome should be unrelated to $\mathcal{C}$ (Foulds & Pan, 2020), it can be unfair if important risk factors for the outcome are associated with demographics (Hardt et al., 2016). For example, if the underlying rates of an illness vary demographically, requiring demographic parity can result in a healthier patients from one group being admitted more often than patients who urgently need care.

## A.2 EXTENDED THEORETICAL ANALYSIS

**Illustrating Relationships between Definitions** Fig. 3 shows how the definitions of MC, DC, and PMC are related. In each subplot, the x and y coordinates map the guarantee from one metric (x axis) to the implied guarantee in the other metric (y axis).

The right panel of Fig. 3 illustrates this relation in comparison to the DC-MC relationship described in Appendix A.3, Theorem 4. At small values of $\varepsilon$ and $\alpha$ and when the model is perfectly calibrated overall, $\alpha$-PMC and $\varepsilon$-DC behave similarly. However, given $\delta > 0$, $\varepsilon$-differentially calibrated models suffer from higher MC error than proportionally calibrated models when $\alpha$-PMC $< 0.3$. The right graph also illustrates the feasible range of $\alpha$ for $\alpha$-PMC is $0 < \alpha < 0.5$, past which it does not provide meaningful $\alpha$-MC. The steeper relation between $\alpha$-PMC and MC may have advantages or disadvantages, depending on context. It suggests that, by optimizing for $\alpha$-PMC, small improvements to this measure can result in relatively large improvements to MC; conversely, $\varepsilon$-DC models that are well calibrated may satisfy a lower value of $\alpha$-MC over a larger range of $\varepsilon$.

### A.2.1 DISCRETIZATION

To clarify and simplify our analysis, we work mainly with the continuous versions of multicalibration and proportional multicalibration, under the assumption that minimizing the discretized versions (i.e., binning $R(x)$) will translate to low values of the continous version. In this section we provide detailed bounds on the continuous versions of PMC and DC that are implied by the discretized versions.

First, we will formally define two different discretization schemes. The first, $\lambda$-discretization, defines equally spaced bins on the interval $[0, 1]$, as follows.

**Definition 10** ($\lambda$-discretization.). *Let $\lambda \in [0, 1], \rho \in [0, 1]$. The $\lambda$-discretization of $[0, 1]$ is denoted by a set of intervals, $\Lambda_\lambda = \left\{ \{I_j\}_{j=0}^{1/\lambda - 1} \right\}$, where $I_j = [j\lambda, (j + 1)\lambda)$.*

For ensuring multiplicative closeness under PMC, it can be useful to instead discretize the prediction bins so that the bins are equally spaced on a log scale. We define such a discretization below.

**Definition 11** (($\lambda, \rho$)-geometric discretization.). *Let $\lambda \in [0, 1], \rho \in [0, 1]$. The $(\lambda, \rho)$-geometric discretization of $[0, 1]$ is denoted by a set of intervals, $\Lambda_\lambda^\rho = \left\{ \{I_j\}_{j=0}^{1/\lambda - 1} \right\}$, where $I_j = [\rho^{(1-j\lambda)}, \rho^{(1-j\lambda-\lambda)})$.*

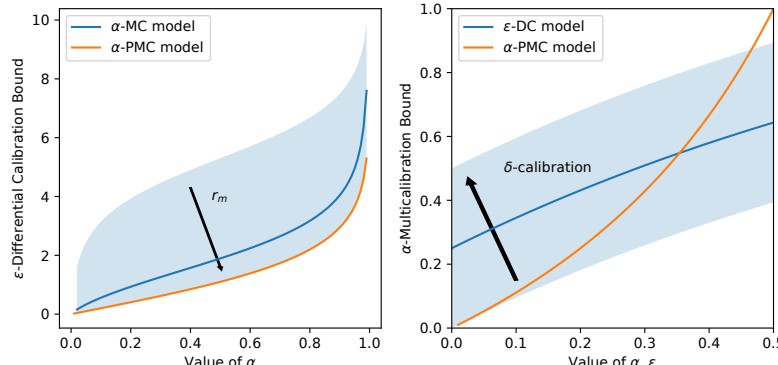

Figure 3: A comparison of $\varepsilon$-DC, $\alpha$-MC, and $\alpha$-PMC in terms of their parameters $\alpha$ and $\varepsilon$. In both panes, the x value is a given value of one metric for a model, and the y axis is the implied value of the other metric, according to Theorem 4-Theorem 3. The left filled area denotes the dependence of the privacy/DC of $\alpha$-multicalibrated models on the minimum risk interval, $r_{min} \in [0.01, 1.0]$. The right filled area denotes the dependence of the MC of $\varepsilon$-differentially calibrated models on their overall calibration, $\delta \in [0.0, 0.5]$. $\alpha$-PMC does not have these sensitivities.

Hebert-Johnson et al. (2018) define a discretized version of MC in which $R(x)$ is binned according to a discretization parameter, $\lambda$:

**Definition 12** (($\alpha, \lambda$)-multicalibration). *Let $\mathcal{C} \subseteq 2^{\mathcal{X}}$ be a collection of subsets of $\mathcal{X}$. For any $\alpha, \lambda > 0$, a predictor $R$ is ($\alpha, \lambda$)-multicalibrated on $\mathcal{C}$ if, for all $I \in \Lambda_\lambda$ and $S \in \mathcal{C}$ where $P_D(R \in I | x \in S) \geq \alpha\lambda$,*

$$\left| \mathbb{E}_D[y | R \in I, x \in S] - \mathbb{E}_D[R | R \in I, x \in S] \right| \leq \alpha.$$

Hébert-Johnson et al. (2018) establish that ($\alpha, \lambda$)-multicalibrated models are at most ($\alpha + \lambda$)-multicalibrated.

In an analogous fashion, we show below that ($\alpha, \lambda$)-PMC implies ($\alpha + \lambda/\rho$)-PMC for bins defined by a $\lambda$-discretization. When using a ($\lambda, \rho$)-geometric discretization, ($\alpha, \lambda$)-PMC implies ($\alpha\rho^{-\lambda} + \rho^{-\lambda} - 1$)-PMC, which can be a tighter bound than the former.

**Claim 1.** *Define $\rho, \alpha, \lambda > 0$ and let $\mathcal{C} \subseteq 2^{\mathcal{X}}$ be a collection of subsets of $\mathcal{X}$. Let $\mathbb{E}_D[y | R \in I, x \in S] \geq \rho$ for all $S \in \mathcal{C}$ and $I \in \Lambda_\lambda$. Let $R(x)$ be a model satisfying ($\alpha, \lambda$)-proportional multicalibration. Then $R(x)$ is at most ($\alpha + \frac{\lambda}{\rho}$)-proportionally multicalibrated.*

*Proof.* By Definition 6, $R(x)$ satisfies

$$\frac{|\mathbb{E}_D[y | R \in I, x \in S] - \mathbb{E}_D[R | R \in I, x \in S]|}{\mathbb{E}_D[y | R \in I, x \in S]} \leq \alpha$$

for categories $(S, I) \in \mathcal{C} \times \Lambda_\lambda$ satisfying $P_D(R(x) \in I | x \in S) \geq \alpha\lambda$. Given $1/\lambda$ bins, the subset where $P_D(R(x) \in I | x \in S) < \alpha\lambda$ has a size of at most $\alpha|S|$. Therefore there is a subset $|S'| \geq (1 - \alpha)|S|$ where for all $r \in \Lambda_\lambda$, $\alpha$-PMC (Definition 5) is satisfied.

Let $\delta$ be the constaint on $\delta$-PMC. Let $p^* = \mathbb{E}_D[y | R = r, x \in S]$ and $r = \mathbb{E}_D[R | R = r, x \in S]$. Consider the case $r > p^*$ and let $\alpha = (r - p^*)/p^*$. $\lambda$-discretization shifts $r$ by at most $\lambda$. Let

$$\delta \leq (r + \lambda - p^*)/p^*$$

Substituting $r \leq \alpha p^* + p^*$ yields

$$\delta \leq \alpha + \frac{\lambda}{p^*}$$

Plugging in $\rho$ as the minimum of $p^*$, we complete the proof.

$\square$ $\square$

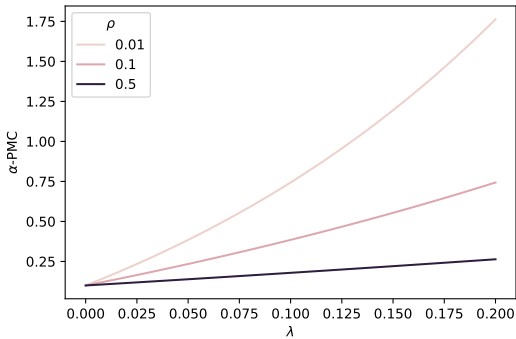

Figure 4: Relationship between $(\alpha, \lambda)$-PMC and $\alpha$-PMC given a geometric discretization. Illustrated for $(\alpha, \lambda) - PMC = 0.1$, for various values of $\rho$ and $\lambda$.

The term $\frac{\lambda}{\rho}$ can be potentially large when $\rho < \lambda$. One way to avoid this issue is to make the change in $R(x)$ between bins scale with $R(x)$ using Definition 11. What makes Definition 11 different from $\lambda$-discretization is that the intervals are a multiplicative, rather than additive, distance apart. Hence, for a given $r \in [0, 1]$, a model satisfying $(\alpha, \lambda)$-PMC can have its prediction shift by at most a factor of $\rho^{-\lambda}$. This leads us to the following proposition.

**Claim 2.** *Define $\rho, \alpha, \lambda > 0$ and let $\mathcal{C} \subseteq 2^{\mathcal{X}}$ be a collection of subsets of $\mathcal{X}$. Let $\mathbb{E}_D[y|R \in I, x \in S] \geq \rho$ for all $S \in \mathcal{C}$ and $I \in \Lambda_\lambda$. Let $R(x)$ be a model satisfying $(\alpha, \lambda)$-proportional multicalibration. Given a $(\lambda, \rho)$-geometric discretization, $R(x)$ is at most $(\alpha\rho^{-\lambda} + \rho^{-\lambda} - 1)$-proportionally multicalibrated.*

*Proof.* By Definition 6, $R(x)$ satisfies

$$\frac{|\mathbb{E}_D[y|R \in I, x \in S] - \mathbb{E}_D[R|R \in I, x \in S]|}{\mathbb{E}_D[y|R \in I, x \in S]} \leq \alpha$$

for categories $(S, I) \in \mathcal{C} \times \Lambda_\lambda$ satisfying $P_D(R(x) \in I|x \in S) \geq \alpha\lambda$. Given $1/\lambda$ bins, the subset where $P_D(R(x) \in I|x \in S) < \alpha\lambda$ has a size of at most $\alpha|S|$. Therefore there is a subset $|S'| \geq (1 - \alpha)|S|$ where for all $r \in \Lambda_\lambda$, $\alpha$-PMC (Definition 5) is satisfied.

Let $\delta$ be the constaint on $\delta$-PMC. Let $p^* = \mathbb{E}_D[y|R = r, x \in S]$ and $r = \mathbb{E}_D[R|R = r, x \in S]$. Consider the case $r > p^*$ and let $\alpha = (r - p^*)/p^*$, i.e. the tight bound. $(\lambda, \rho)$-geometric discretization shifts $r$ by at most a factor of $\rho^{-\lambda}$. This implies

$$\delta \leq (r\rho^{-\lambda} - p^*)/p^*$$

Substituting $r = \alpha p^* + p^*$ yields

$$\delta \leq \alpha\rho^{-\lambda} + \rho^{-\lambda} - 1.$$

$\square$ $\square$

We illustrate the relationship between $(\alpha, \lambda)$-PMC and $\alpha$-PMC given a geometric discretization in Fig. 4, which quantifies the relationship for different settings of $\lambda$ and $\rho$.

### A.2.2 ADDITIONAL DEFINITIONS

**Definition 13** (($\varepsilon, \lambda$)-Differential Calibration). *Let $\mathcal{C} \subseteq 2^{\mathcal{X}}$ be a collection of subsets of $\mathcal{X}$. A model $R(x)$ is ($\varepsilon, \lambda$)-differentially calibrated with respect to $\mathcal{C}$ if, for all pairs $(S_i, S_j) \in \mathcal{C} \times \mathcal{C}$ for which $P_D(S_i), P_D(S_j) > 0$, for any $I \in \Lambda_\lambda$,*

$$e^{-\varepsilon} \leq \frac{\mathbb{E}_D[y|R \in I, x \in S_i]}{\mathbb{E}_D[y|R \in I, x \in S_j]} \leq e^{\varepsilon} \tag{6}$$

### A.2.3 LOSS FUNCTIONS

The following loss functions are empirical analogs of the definitions of $MC$, $PMC$, and $DC$, and are used in the experiment section to measure performance.

**Definition 14** (MC loss). *Let $\mathcal{D} = \{(y,x)_i\}_{i=0}^N \sim D$, and let $\alpha, \lambda, \gamma > 0$. Define a collection of subsets $\mathcal{C} \in 2^{\mathcal{X}}$ such that for all $S \in \mathcal{C}, |S| \geq \gamma N$. Let $S_I = \{x : R(x) \in I, x \in S\}$ for $(S,I) \in \mathcal{C} \times \Lambda_\lambda$. Define the collection $\mathcal{S}$ containing all $S_I$ satisfying $S_I \geq \alpha \lambda N$. The MC loss of a model $R(x)$ on $\mathcal{D}$ is*

$$\max_{S_I \in \mathcal{S}} \frac{1}{|S_I|} \left| \sum_{i \in S_I} y_i - \sum_{i \in S_I} R_i \right|$$

**Definition 15** (PMC loss). *Let $\mathcal{D} = \{(y,x)_i\}_{i=0}^N \sim D$, and let $\alpha, \lambda, \gamma, \rho > 0$. Define a collection of subsets $\mathcal{C} \in 2^{\mathcal{X}}$ such that for all $S \in \mathcal{C}, |S| \geq \gamma N$. Let $S_I = \{x : R(x) \in I, x \in S\}$ for $(S,I) \in \mathcal{C} \times \Lambda_\lambda$. Define the collection $\mathcal{S}$ containing all $S_I$ satisfying $S_I \geq \alpha \lambda N$. Let $\frac{1}{|S_I|} \sum_{i \in S_I} y_i \geq \rho$. The PMC loss of a model $R(x)$ on $\mathcal{D}$ is*

$$\max_{S_I \in \mathcal{S}} \frac{\left| \sum_{i \in S_I} y_i - \sum_{i \in S_I} R_i \right|}{\sum_{i \in S_I} y_i}$$

**Definition 16** (DC loss). *Let $\mathcal{D} = \{(y,x)_i\}_{i=0}^N \sim D$, and let $\alpha, \lambda, \gamma > 0$. Define a collection of subsets $\mathcal{C} \in 2^{\mathcal{X}}$ such that for all $S \in \mathcal{C}, |S| \geq \gamma N$. Given a risk model $R(x)$ and prediction intervals $I$, Let $S_I = \{x : R(x) \in I, x \in S\}$ for $(S,I) \in \mathcal{C} \times \Lambda_\lambda$. Define the collection $\mathcal{S}$ containing all $S_I$ satisfying $S_I \geq \alpha \lambda N$. The DC loss of a model $R(x)$ on $\mathcal{D}$ is*

$$\max_{(S_I^a, S_I^b) \in \mathcal{S} \times \mathcal{S}} \log \left| \frac{1}{|S_I^a|} \sum_{i \in S_I^a} y_i - \frac{1}{|S_I^b|} \sum_{j \in S_I^b} y_j \right|$$

### A.2.4 THEOREM PROOFS

**Theorem 3** *Let $R(x)$ be a model satisfying $\alpha$-PMC on a collection $\mathcal{C}$. Then $R(x)$ is $(\frac{\alpha}{1-\alpha})$-multicalibrated on $\mathcal{C}$.*

*Proof.* To distinguish the parameters, let $R(x)$ be a model satisfying $\delta$-PMC. Let $r = \mathbb{E}_D[R|R = r, x \in S]$ and $p^* = \mathbb{E}_D[y|R = r, x \in S]$. Then $r/(1+\delta) \leq p^* \leq r/(1-\delta)$. We solve for the upper bound on $\alpha$-MC from Definition 2 for the case when $p^* > r$. This yields

$$\begin{aligned}
\alpha &\leq p^* - r \\
&\leq \frac{r}{1-\delta} - r \\
&= r\frac{\delta}{1-\delta} \\
&\leq \frac{\delta}{1-\delta}.
\end{aligned}$$

We can also solve for the lower bound on $\alpha$-MC from Definition 2 for the case when $p^* < r$. This yields

$$\begin{aligned}
\alpha &\leq r - p^* \\
&\leq r - \frac{r}{1+\delta} \\
&= r\frac{\delta}{1+\delta} \\
&\leq \frac{\delta}{1+\delta}.
\end{aligned}$$

For any $\delta > 0$, $\frac{\delta}{1-\delta} > \frac{\delta}{1+\delta}$. Therefore the first case ($p^* > r$) limits the multicalibration of $R(x)$.

$\square$ $\square$

**Proposition 1** *Define $\alpha, \lambda, \gamma, \rho > 0$. Let $\mathcal{C} \subseteq 2^{\mathcal{X}}$ be a collection of subsets of $\mathcal{X}$ such that, for all $S \in \mathcal{C}$, $P_D(S) > \gamma$. Let $R(x)$ be a risk prediction model to be post-processed. For all $(S, I) \in \mathcal{C} \times \Lambda_\lambda$, let $E[y|R \in I, x \in S] > \rho$. There exists an algorithm that satisfies $(\alpha, \lambda)$-PMC with respect to $\mathcal{C}$ in $O(\frac{|C|}{\alpha^3 \lambda^2 \rho^2 \gamma})$ steps.*

*Proof.* We show that Algorithm 1 converges using a potential function argument (Bansal & Gupta, 2019), similar to the proof techniques for the MC boosting algorithms in Hebert-Johnson et al. (2018); Kim et al. (2019). Let $p_i^*$ be the underlying risk, $R_i$ be our initial model, and $R_i'$ be our updated prediction model for individual $i \in S_r$, where $S_r = \{x|x \in S, R(x) \in I\}$ and $(S, I) \in \mathcal{C} \times \Lambda_\lambda$. We use $p^*$, $R$, and $R'$ without subscripts to denote these values over $S_r$. We cannot easily construct a potential argument using progress towards $(\alpha, \lambda)$-PMC, since its derivative is undefined at $\mathbb{E}_D[y|R \in I, x \in S]=0$. Instead, we analyze progress towards the difference in the $\ell_2$ norm at each step.

$$
\begin{aligned}
||p^* - R|| - ||p^* - R'|| &= \sum_{i \in S_r} (p_i^* - R_i)^2 - \sum_{i \in S_r} (p_i^* - \text{squash}(R_i + \Delta r))^2 \\
&\geq \sum_{i \in S_r} \left((p_i^* - R)^2 - (p_i^* - (R_i + \Delta r))^2\right) \\
&= \sum_{i \in S_r} \left(2p_i^* \Delta r - 2R_i \Delta r - \Delta r^2\right) \\
&= 2\Delta r \sum_{i \in S_r} (p_i^* - R_i) - |S_r|\Delta r^2 \quad (7)
\end{aligned}
$$

From Algorithm 1 we have

$$
\Delta r = \frac{1}{|S_r|} \sum_{i \in S_r} (p_i^* - R_i)
$$

Substituting into Eq. (7) gives

$$
||p^* - R|| - ||p^* - R'|| \geq |S_r|\Delta r^2
$$

We know that $|S_r| \geq \alpha\lambda\gamma N$, and that the smallest update $\Delta r$ is $\alpha\rho$. Thus,

$$
||p^* - R|| - ||p^* - R'|| \geq \alpha^3 \rho^2 \lambda \gamma N
$$

Since our initial loss, $||p^* - R||$, is at most $N$, Algorithm 1 converges in at most $O(\frac{1}{\lambda^3 \rho^2 \lambda \gamma})$ updates for category $S_r$.

To understand the total number of steps, including those without updates, we consider the worst case, in which only a single category $S_r$ is updated in a cycle of the for loop (if no updates are made, the algorithm exits). Since each repeat consists of at most $|C|/\lambda$ loop iterations, this results in $O(\frac{|C|}{\alpha^3 \lambda^2 \rho^2 \gamma})$ total steps. □ □

### A.3 ADDITIONAL THEOREMS

#### A.3.1 DIFFERENTIALLY CALIBRATED MODELS WITH GLOBAL CALIBRATION ARE MULTICALIBRATED

Here we show that, under the assumption that a model is globally calibrated (satisfies $\delta$-calibration), models satisfying $\varepsilon$-DC are also multicalibrated.

**Theorem 4.** *Let $R(x)$ be a model satisfying $(\varepsilon, \lambda)$-DC and $\delta$-calibration. Then $R(x)$ is $(1 - e^{-\varepsilon} + \delta, \lambda)$-multicalibrated.*

*Proof.* From Eq. (2) we observe that $\varepsilon$ is bounded by the two groups with the largest and smallest group- and prediction- specific probabilities of the outcome. Let $I_M$ be the risk stratum maximizing $(\varepsilon, \lambda)$-DC, and let $p_n = \max_{S \in \mathcal{C}} P_D(y | R \in I_M, x \in S)$ and $p_d = \min_{S \in \mathcal{C}} P_D(y | R \in I_M, x \in S)$. These groups determine the upper and lower bounds of $\varepsilon$ as $e^{-\varepsilon} \leq p_d / p_n$ and $p_n / p_d \leq e^{\varepsilon}$.

We note that $p_d \leq P_D(y | R \in I_M) \leq p_n$, since $P(y | R \in I_M) = \frac{1}{N} \sum_{S \in \mathcal{C}} |S| P_D(y | R \in I_M, x \in S)$, and $p_n$ and $p_d$ are the extreme values of $P(y | R \in I_M, x \in S)$ among $S$. So, $\alpha$-MC is bound by the group outcome that most deviates from the predicted value, which is either $p_n$ or $p_d$. Let $r = P_D(R | R \in I_M)$. There are then two scenarios to consider:

1. $\alpha \leq |p_n - r| = p_n - r$ when $r \leq \frac{1}{2}(p_n + p_d)$; and

2. $\alpha \leq |p_d - r| = r - p_d$ when $r \geq \frac{1}{2}(p_n + p_d)$.

We will look at the first case. Let $p_r^* = P_D(y | R \in I_M)$. Due to $\delta$-calibration, $p_r^* - \delta \leq r \leq p_r^* + \delta$. Then

$$
\begin{aligned}
\alpha &\leq p_n - r \\
&\leq p_n - (p_r^* - \delta) \\
&\leq p_n - p_d + \delta \\
&= p_n(1 - e^{-\varepsilon}) + \delta \\
\alpha &\leq 1 - e^{-\varepsilon} + \delta.
\end{aligned}
$$

Above we have used the facts that $r \leq p_r^* - \delta$, $p_r^* \geq p_d$, $p_d \leq e^{-\varepsilon} p_n$, and $p_n \leq 1$. The second scenario is complementary and produces the identical bound. □ □

Theorem 4 formally describes how $\delta$-calibration controls the baseline calibration error contribution to $\alpha$-MC, while $\varepsilon$-DC limits the deviation around this value by constraining the (log) maximum and minimum risk within each category.

### A.4 MULTICALIBRATED MODELS SATISFY INTERSECTIONAL GUARANTEES

In contrast to DF, MC (Hebert-Johnson et al., 2018) was not designed to explicitly incorporate the principles of intersectionality. However, we show that it provides an identical efficiency property to DF in the theorem below. Given an individual's attributes $x = (x_1, \ldots, x_d)$, it will be useful to refer to subsets we wish to protect, e.g. demographic identifiers. To do so, we define $\mathcal{A} = \{A_1, \ldots, A_p\}$, $p \leq d$, such that $A_1$ is the set of values taken by attribute $x_1$.

**Theorem 5.** *Let the collection of subsets $\mathcal{C} \subseteq 2^{\mathcal{X}}$ define groups of individuals according to the Cartesian product of attributes $A \subseteq \mathcal{A}$. Let $\mathcal{G} \in 2^{\mathcal{X}}$ be any collection of subsets that groups individuals by the Cartesian product of attributes in $A'$, where $A' \subset A$ and $A' \neq \emptyset$. If $R(x)$ satisfies $\alpha$-MC on $\mathcal{C}$, then $R(x)$ is $\alpha$-multicalibrated on $\mathcal{G}$.*

In proving Theorem 5, we will make use of the following lemma.

**Lemma 6.** *The $\alpha$-MC criteria can be rewritten as: for a collection of subsets $\mathcal{C} \subseteq \mathcal{X}$, $\alpha \in [0, 1]$, and $r \in [0, 1]$,*

$$
\max_{c \in \mathcal{C}} \mathbb{E}_D[y | R(x) = r, x \in c] \leq r + \alpha
$$

*and*

$$
\min_{c \in \mathcal{C}} \mathbb{E}_D[y | R(x) = r, x \in c] \geq r - \alpha
$$

*Proof.* The lemma follows from Definition 2, and simply restates it as a constraint on the maximum and minimum expected risk among groups at each prediction level. □ □

*Proof of Theorem 5.* We use the same argument as Foulds et al. (2019a) in proving this property for DF. Define $Q$ as the Cartesian product of the protected attributes included in $\mathcal{A}$, but not $\mathcal{A}'$. Then for any $(y, x) \sim D$,

$$\max_{g \in \mathcal{G}} \mathbb{E}_D[y|R(x) = r, x \in g] = \max_{g \in \mathcal{G}} \sum_{q \in Q} \mathbb{E}_D[y|R(x) = r, x \in g \cap q] P[x \in q|x \in g] \tag{8}$$

$$\leq \max_{g \in \mathcal{G}} \sum_{q \in Q} \max_{q' \in Q} \mathbb{E}_D[y|R(x) = r, x \in g \cap q'] P[x \in q|x \in g] \tag{9}$$

$$= \max_{g \in \mathcal{G}} \max_{q' \in Q} \mathbb{E}_D[y|R(x) = r, x \in g \cap q'] \tag{10}$$

$$= \max_{c \in \mathcal{C}} \mathbb{E}_D[y|R(x) = r, x \in c]. \tag{11}$$

Moving from (5) to (6) follows from substituting the maximum value of $\mathbb{E}_D[y|R(x) = r, x]$ for observations in the intersection of subsets in $\mathcal{G}$ and $Q$ which is the upper limit of the expression in (5). Moving from (6) to (7) follows from recognizing that the sum $P[x \in q|x \in g]$ for all subsets in $\mathcal{Q}$ is 1. Finally, moving from (7) to (8) follows from recognizing that the intersections of subsets in $\mathcal{G}$ and $\mathcal{Q}$ that satisfy (7), must define a subset of $\mathcal{C}$. Applying the same argument, we can show that

$$\min_{g \in \mathcal{G}} \mathbb{E}_D[y|R(x) = r, x \in g] \geq \min_{c \in \mathcal{C}} \mathbb{E}_D[y|R(x) = r, x \in c].$$

Substituting into Lemma 6,

$$\max_{g \in \mathcal{G}} \mathbb{E}_D[y|R(x) = r, x \in g] \leq \alpha + r$$

and

$$\min_{g \in \mathcal{G}} \mathbb{E}_D[y|R(x) = r, x \in g] \geq r - \alpha$$

or

$$\left| \mathbb{E}_D[y|R(x) = r, x \in g] - r \right| \leq \alpha$$

for all $g \in \mathcal{G}$. Therefore $R(x)$ is $\alpha$-multicalibrated with respect to $\mathcal{G}$.

□                                                                                          □

As a concrete example, imagine we have the protected attributes $A = \{\text{race} \in \{B, W\}, \text{gender} \in \{M, F\}\}$. According to Theorem 5, $\mathcal{C}$ would contain four sets: $\{(B, M), (B, F), (W, M), (W, F)\}$. In contrast, there are eight possible sets in $\mathcal{G}$: $\{(B, M), (B, F), (W, M), (W, F), (B, *), (W, *), (*, M), (*, F)\}$, where the wildcard indicates a match to either attribute. As noted in Appendix A.1.1, the efficiency property is useful because the number of possible sets in $\mathcal{G}$ grows at a large combinatorial rate, rate as additional attributes are added; meanwhile $\mathcal{C}$ grows at a slower, yet exponential, rate. For an intuition for why this property holds, consider that the maximum calibration error of two subgroups is at least as large as the maximum expected error of those groups combined; e.g., the maximum calibration error in a higher order groups such as $(B, *)$ will be covered by the maximum calibration error in either $(B, M)$ or $(B, F)$.

## A.5   ADDITIONAL EXPERIMENT DETAILS

Models were trained on a heterogenous computing cluster. Each training instance was limited to a single core and 4 GB of RAM. We conducted a full parameter sweep of the parameters specified in Table 2. A single trial consisted of a method, a parameter setting from Table 2, and a random seed. Over 100 random seeds, the data was shuffled and split 75%/25% into train/test sets. Results in the manuscript are summarized over these test sets.

**Code** Code for the experiments is available here: https://github.com/by1tTZ4IsQkAO80F/pmc. Code is licensed under GNU Public License v3.0.

Table 5: Features used in the hospital admission task.

| Description | Features |
|---|---|
| Vitals | temperature, heartrate, resprate, o2sat, systolic blood pressure, diastolic blood press, |
| Triage Acuity | Emergency Severity Index (Tanabe et al., 2004) |
| Check-in Data | chief complaint, self-reported pain score |
| Health Record Data | no. previous visits, no. previous admissions |
| Demographic Data | ethnoracial group, gender, age, marital status, insurance, primary language |

**Data** We make use of data from the MIMIC-IV-ED repository, version 1.0, to train admission risk prediction models (Johnson et al., 2021). This resource contains more than 440,000 ED admissions from Beth Isreal Deaconness Medical Center between 2011 and 2019. We preprocessed these data to construct an admission prediction task in which our model delivers a risk of admission estimate for each ED visitor after their first visit to triage, during which vitals are taken. Additional historical data for the patient was also included (e.g., number of previous visits and admissions). A list of features is given in Table 5.

### A.6 ADDITIONAL RESULTS

Table 2 lists a few parameters that may affect the performance of post-processing for both MC and PMC. Of particular interest when comparing MC versus PMC post-processing is the parameter $\alpha$, which controls how stringent the calibration error must be across categories to terminate, and the group definition ($A$), which selects which features of the data will be used to asses and optimize fairness. We look at the performance of MC and PMC postprocessing over values of $\alpha$ and group definitions in Figs. 5 to 7. Finally, we empirically compare MC- and PMC-postprocessing by the number of steps required for each to reach their best performance in Fig. 8 and Table 4.

From Fig. 5, it is clear that post-processing has a minimal effect on AUROC in all cases; note the differences disappear if we round to two decimal places. When post processing with RF, we do note a relationship between lower values of $\alpha$ and a very slight decrease in performance, particularly for MC-postprocessing.

Figs. 6 and 7 show performance between methods on MC loss and PMC loss, respectively. In terms of MC loss, PMC-postprocessing tends to produce models with the lowest loss, at $\alpha$ values greater than 0.01. Lower values of $\alpha$ do not help MC-postprocessing in most cases, suggesting that these smaller updates may be overfitting to the post-processing data. In terms of PMC loss (Fig. 7), we observe that performance by MC-postprocessing is highly sensitive to the value of $\alpha$. For smaller values of $\alpha$, MC-postprocessing is able to achieve decent performance by these metrics, although in all cases, PMC-postprocessing generates a model with a better median loss value at some configuration of $\alpha$.

We assess how many steps/updates MC and PMC take for different values of $\alpha$ in Fig. 8, and summarize empirical measures of running time in Table 4. On the figure, we annotate the point for which each post-processing algorithm achieves the lowest median value of PMC loss across trials. Fig. 8 validates that PMC-postprocessing is more efficient than MC-postprocessing at producing models with low PMC loss, on average requiring 4.0x fewer updates to achieve its lowest loss on test. From Table 4 we observe that PMC typically requires a larger number of updates to achieve its best performance on MC loss (about 2x wall clock time and number of updates), whereas MC-postprocessing requires a larger number of updates to achieves its best performance on PMC loss and DC loss, due to its dependence on very small values of $\alpha$. We accompany these results with the caveat that they are based on performance on one real-world task, and wall clock time measurements are influenced by the heterogenous cluster environment; future work could focus on a larger empirical comparison.

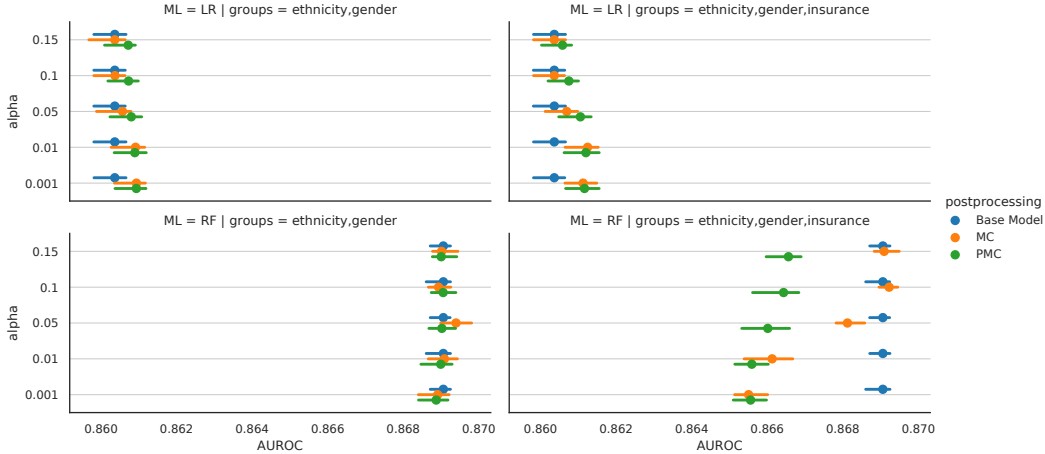

Figure 5: AUROC test performance versus $\alpha$ across experiment settings. Rows are different ML base models, and columns are different attributes used to define $\mathcal{C}$. The color denotes the post-processing method.

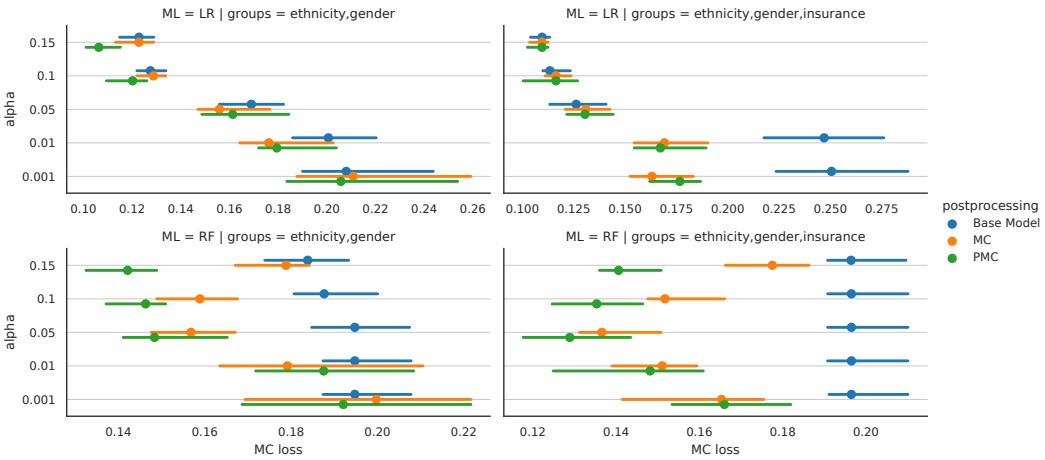

Figure 6: MC loss test performance versus $\alpha$ across experiment settings. Rows are different ML base models, and columns are different attributes used to define $\mathcal{C}$. The color denotes the post-processing method.

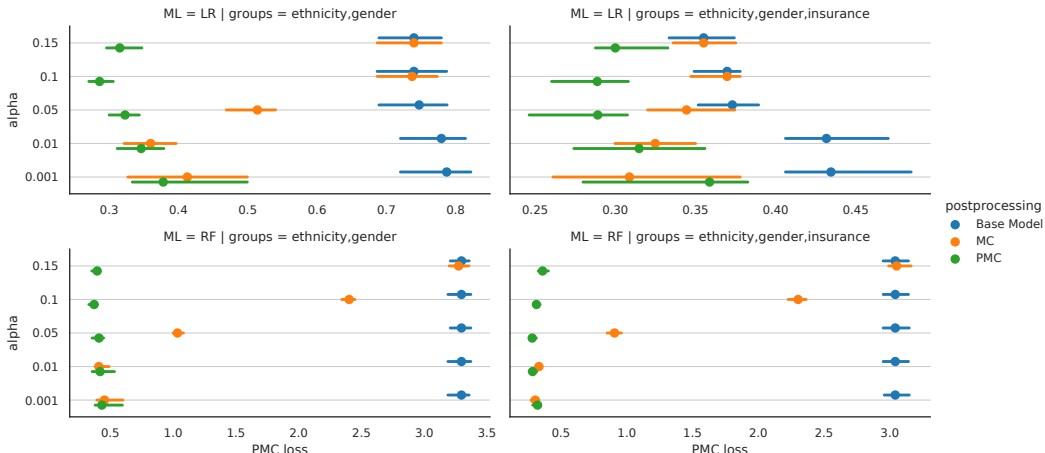

Figure 7: PMC loss test performance versus $\alpha$ across experiment settings. Rows are different ML base models, and columns are different attributes used to define $\mathcal{C}$. The color denotes the post-processing method.

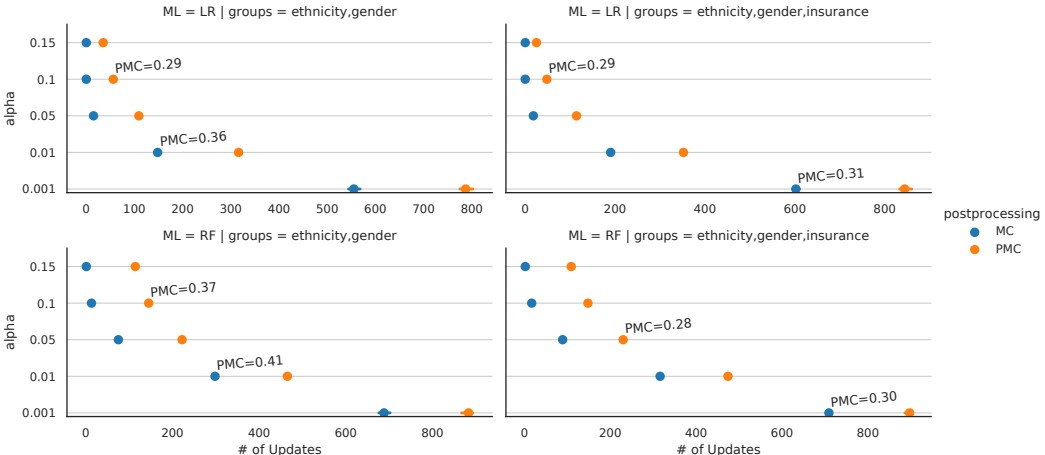

Figure 8: Number of post-processing updates by MC and PMC versus $\alpha$ across experiment settings. Rows are different ML base models, and columns are different attributes used to define $\mathcal{C}$. The color denotes the post-processing method. Each result is annotated with the median PMC loss for that method and parameter combination.

