# OpenReview forum: "Proportional Multicalibration"
_ICLR.cc/2023/Conference — Submitted to ICLR 2023_

### Official Review · Reviewer_PsRB · 2022-10-17

**Confidence:** 4
**Correctness:** 2
**Technical Novelty And Significance:** 3
**Empirical Novelty And Significance:** 3
**Recommendation:** 5

**Clarity, Quality, Novelty And Reproducibility:**

Discarding the mistakes, the work is high quality and novel.

Clarity:

Section 2.1. "Given an individual’s attributes $x = (x_1, . . . , x_d)$, it will be useful to refer to subsets we wish to protect, e.g. demographic identifiers." It is unclear what the authors want to say.

Section 2.1. "To do so, we define $A = \{A_1, . . . , A_p\}$, $p \le d$, such that $A_1 = \{x_{1i}
, . . . , x_{1k}\}$ is a finite set of values taken by attribute $x_1$." First of all, I do not understand what the authors want to say here (what are i and k? for example), second this notation is never used in this work.

Section 2.3, second line. "for all $S \in C$" should be "for all $S \in C$ and $r \in [0, 1]$"

In the last sentence before section 2.4, the references should be inside parentheses.



**Strength And Weaknesses:**

Strengths:

This work is clearly written and addresses an important problem, calibration, that is relevant to model performance evaluation, trustworthiness, and algorithmic fairness. It is a blend of theoretical and computational contributions. The work is novel and provides new insight into the relationship between different definitions of calibration and addresses some of their limitations. The figures are very clear, and professional, and confidence intervals are computed. In general a strong contribution to the ICLR conference.

Weaknesses:
While due to limited time, I could not go through all the derivations and proofs in the appendices, and I found several mistakes in the proofs provided in the main text and computations. I might have made a mistake, but if I am not mistaken this work need to be reviewed comprehensively to make sure that there are no other mistakes in the proofs.


Major issues::

1. "Proportionally multicalibrated models thereby obtain robust fairness guarantees that are independent of population risk categories." This is actually not strictly correct. In the proof of Theorem 3, we have $=\frac{r\delta}{1-\delta} \le \frac{\delta}{1-\delta}$. While this is not incorrect, this is not the strongest upper bound, one can alternatively write $=\frac{r\delta}{1-\delta} \le \frac{r_{max} \delta}{1-\delta}$ that is both stronger than the claim made in Theorem 3 and makes this bound dependent on population risk categories. Hence, under $\alpha$-PMC, error in MC can depend on the population risk category.

2. Theorem 1.
After  plugging the lower and upper bounds into Eq. (2), we get $e^{\epsilon} \le \frac{r_i + \alpha}{r_j - \alpha} $ not $e^{\epsilon} \ge \frac{r + \alpha}{r - \alpha}$. Therefore, the maximum of this ratio will be  ${\epsilon}  \le \ln(\frac{r_{\rm max} + \alpha}{r_{\rm min} - \alpha})$  not ${\epsilon} \le \log(\frac{r_{\rm min} + \alpha}{r_{\rm min} - \alpha})$. Please also use $\ln$ as $\log$ is often reserved for log-10 based. Also, the inequalities should be $ \le$, not $\ge$. If it is $\ge$ then $\epsilon$ can be any large number and does not provide a meaningful bound on differential calibration.

3. In a similar vein, in Theorem 2, the r in the denominator and nominator can be different and do not need to be the same, hence they do not cancel and the bound is still a function of $r$.

4. I cannot replicate the numbers $6$% and $20$%, neither with the new formula nor with the old formula. Can you walk me through how these numbers are computed?


Minor Issues:

4. In Figure 3 (and results), the results are sensitive to the choice of $\alpha$, and as the authors mentioned the optimal choice of $\alpha$ might differ. I could not understand how the $\alpha$ is chosen for each model (MC and MPC).





**Summary Of The Paper:**

This work identifies a limitation of multicalibration, relates multicalibration and differential calibration, and proposes a new metric of calibration, dubbed proportional multicalibration (PMC). Then it proposes an algorithm to achieve PMC and illustrate the performance of the proposed algorithm in simulated and hospital admission settings.

**Summary Of The Review:**

While this paper is interesting and the contribution is novel and important, the current work requires another major review to make sure the proofs and math work out and there are no mistakes. Therefore, I recommend rejection.

If I misunderstood the proofs, please correct me. I would be glad to change my score.

---

> ### Author Response · Authors · 2022-11-14
> **response to PsRB: fixed mistakes and clarification**
>
> We appreciate the reviewer's time with this manuscript and the careful reading of the details. We believe our updated version addresses the reviewer's concerns and are open to further discussion this week.
>
> ## 1. Independence Statement
>
> > "Proportionally multicalibrated models thereby obtain robust fairness guarantees that are independent of population risk categories." This is actually not strictly correct.
> >
>
> We agree with the reviewer’s point, although the goal is that the value of PMC is independent of population risk, rather than the value of MC, which will always depend on population risk. Our simple solution is to update this line to “Proportionally multicalibrated models thereby obtain robust fairness guarantees that **less** independent of population risk categories."
>
> If this is not satisfactory to the reviewer please let us know and we can revisit.
>
> ## 2. Theorem 1
>
> > *After plugging the lower and upper bounds into Eq. (2), we get $e^{\epsilon} \leq \frac{r_i+\alpha}{r_j + \alpha}$ not $e^{\epsilon} \geq \frac{r+\alpha}{r + \alpha}$. Therefore, the maximum of this ratio will be $e^{\epsilon} \leq \ln(\frac{r_{max}+\alpha}{r_{min} + \alpha})$   not $e^{\epsilon} \leq \log ( \frac{r+\alpha}{r + \alpha})$.*
> >
>
> Thank you for the careful reading. The reviewer is correct that, with the discrete versions of MC and DC, i.e. $(\alpha, \lambda)$-MC and $(\epsilon, \lambda)$-DC, the discretized predictions $r$ depend on the group. This proof was instead meant for connecting $\alpha$-MC to $\epsilon$-DC, rather than . We have fixed this mistake by writing the theorems using the continuous definitions. We have also added text indicating how the discrete and continuous versions of MC, DC and PMC are connected, so that we claim satisfying the discrete case essentially satisfies the continuous one.
>
> > *Please also use ln as log  is often reserved for log-10 based.*
> >
>
> Done.
>
> > *Also, the inequalities should be $\leq$ , not$\leq$ . If it is > then* $\epsilon$ *can be any large number and does not provide a meaningful bound on differential calibration.*
> >
>
> Sorry, we do not this is not correct. $\epsilon$-DC bounds the differential on both sides as
>
> $e^{-\epsilon} \leq \frac{E[y|R,S_i]}{E[y|R,S_j]} \leq e^{\epsilon}$
>
> meaning that, as the reviewer points out, $\epsilon$ can be arbitrarily large and hence meaningless. The goal is therefore to find the **minimum** $\epsilon$ that would **still** be satisfied by an $\alpha$-multicalibrated model. To find it, we focused on the right-hand bound, $e^{\epsilon}$, which constrains the maximum value of the differential. Flipping the left and right sides, this gives us
>
> $e^{\epsilon} \geq \frac{E[y|R,S_i]}{E[y|R,S_j]}$
>
> We then fill in the maximum value of the fraction, leaving
>
> $e^{\epsilon} \geq \frac{r_{min} + \alpha}{r_{min} - \alpha}$
>
> If we pursued the left-hand bound, we could equivalently write
>
> $e^{-\epsilon} \leq \frac{r_{min} - \alpha}{r_{min} + \alpha}$
>
> We’ve added some text to the proof to hopefully make the steps more clear. Perhaps the wording was too vague in the original.
>
> ## 3. Canceling r
>
> See above discussion of discretization
>
> ## 4. Example numbers
>
> > *I cannot replicate the numbers 6% and 20%, neither with the new formula nor with the old formula. Can you walk me through how these numbers are computed?*
> >
>
> Certainly. For r=0.9, an $\alpha$=0.05 means $p^*_{min}$=0.85. As a percent, $(r-p^*_{min})/p^*_{min}$=0.058 or about 6%. Similarly, for $r=0.3, p=0.25$, % error = $(0.3-0.25)/0.25) = 0.2$.
>
> We’ve simplified the example somewhat by stating it with the continuous versions of the metrics (i.e. no intervals).
>
> ## 5. choice of alpha
>
> > *In Figure 3 (and results), the results are sensitive to the choice of $\alpha$, and as the authors mentioned the optimal choice of* $\alpha$  *might differ. I could not understand how the $\alpha$  is chosen for each model (MC and MPC)*
> >
>
> We have swept through 3 orders of magnitude with alpha to try to give a reasonable range for MC or PMC. In the appendix, we break down the results by values of $\alpha$ for both methods. It is clear to us from these figures (5-7) that the range of alphas we have chosen are wide enough to capture an approximate optimal value for either method, including with respect to base ML model and group definition.
>
> ## Clarity
>
> We have addressed each note on clarity. The notation for attributes is used in the Appendix Theorem 6, and we have moved the description of $A$ and $\mathcal{A}$ there.

---

### Official Review · Reviewer_RcCa · 2022-10-23

**Confidence:** 3
**Correctness:** 2
**Technical Novelty And Significance:** 2
**Empirical Novelty And Significance:** 2
**Recommendation:** 3

**Clarity, Quality, Novelty And Reproducibility:**

I find the paper a little lacking in rigour and pedagogy overall. For example:
- I do not understand how the parameters ($\alpha$, …) have been chosen for the experiments on MIMIC (+ cf weaknesses).
- It is unclear to me what is plotted in Figure 1.
- The main paper is not self-contained. The losses used to measure experimental performances are only provided in Appendix. Tables  (eg Table 3) are referred to in the main text without specifying that they are in Appendix.
- Theorem 4 states they there exists an algorithm for proportional multicalibration in a polynomial number of steps, then Algorithm 1 is presented as an algorithm which satisfies Theorem 4. It would sound more natural to me to introduce Algorithm 1, and then have a Proposition which states that it converges in a polynomial number of steps.

In terms of novelty, the proposed measure (proportional multicalibration) is new, and the post-processing algorithm proposed is a direct extension of the one for multicalibration.

Typos:
* efficinet (just before paragraph 1.1
* Outcomes y are random samples from $p^*(y|x)$ instead of $p^*(x)$?
* MC similar on boosting -> MC similar to boosting

**Strength And Weaknesses:**

**Strength**
- The paper adresses a problem related to fairness with important practical implications.
- Completeness: the proposed measure is motivated theoretically, an algorithm is given to implement it practically, and experiments on simulated and real data are provided.

**Weaknessess**
-  I have a major concern regarding the theoretical results (Th. 1, 2, and 3). I am not sure that they are correct.
 Let’s take theorem 1, which states that multicalibration implies differential calibration. In the proof, $r$ is the expected confidence score over a group S in a bin I, and $p^*$ is the expected outcome over a group S in a bin I. In the proof, it seems to me that $p^\star$ and $r$ are considered equal for all bins and groups. However, we have dependencies $r = r(I, S)$ and $p^* = p^*(I, S)$. So plugging into equation (2) would involve the quantities $r(I, S_i)$, $r(I, S_j)$, $p^*(I, S_i)$ and $p^*(I, S_j)$, so that the conclusion does not follow.
Exactly the same comments apply to Theorem 2, which states that proportional multicalibration implies differential calibration.
For Theorem 3, which states that proportional multicalibration implies multicalibration, the proof is done for the case $p^\star > r$. I think the case $p^* < r$ should also be considered, otherwise the proof is not complete and the Theorem could be incorrect.

- It is unclear to me how the hyperparameters have been chosen for the experiments ($\alpha$, $\lambda$, $\gamma$, $\rho$). This is however important to be able to evaluate the experiments correctly.

- The main paper lacks a more thorough discussion/comparison of MC vs PMC in the experimental part. As it seems that MC with small values of $\alpha$ can do as well as PMC for the various performance losses, I understand that in practice, the main advantage of PMC vs MC would be in terms of computational time budget. Do I understand correctly? It would be useful to discuss this in more depth in the main paper.


**Summary Of The Paper:**

The authors introduce the notion of **proportional multicalibration** (PMC): a criteria that measures whether a model is both multicalibrated (MC) and fair (in the sense of differentiable calibration (DC) which is an extension of sufficiency). As a reminder, multicalibration refers to a notion of calibration by group and by bin (of confidence scores). And differential calibration, which is a notion introduced by the authors, ensures that any two groups in a collection have the same expected score for each bin (of confidence scores).

They show theoretically (Th.2 and Th. 3) that PMC implies **both** multicalibration and differential calibration. A post-processing algorithm is then provided to proportionally multicalibrate confidence scores. Finally, experiments on an hospital admission task using MIMIC IV show that the proposed algorithm decreases the MC, DC as well as PMC losses without decreasing the AUROC.


**Summary Of The Review:**

The paper is interesting but penalised by too many unclear points (notably related to the correctness of the theorems, choice of hyperparameters, usefulness of PMC vs MC), that prevent me from being convinced by the paper.

---

> ### Author Response · Authors · 2022-11-14
> **Improved proofs, discussion, clarity**
>
> We thank the reviewer for checking the proofs and reading the paper carefully. Our revised manuscript includes many changes that hopefully address each of the reviewer's criticisms.
>
> > I have a major concern regarding the theoretical results (Th. 1, 2, and 3). I am not sure that they are correct. Let’s take theorem 1, which states that multicalibration implies differential calibration. In the proof, $r$ is the expected confidence score over a group S in a bin I, and $p^*$ is the expected outcome over a group S in a bin I. In the proof, it seems to me that $p^*$ and $r$ are considered equal for all bins and groups. However, we have dependencies $r = r(I,S)$ and* $p=p^*(I,S)$*. So plugging into equation (2) … the conclusion does not follow.
>
> The reviewer (along with reviewer PsRB) correctly pointed out a mistake in the theorems, which is the following: the proofs were written for the continuous versions of MC, PMC, and DC. If we restate the theorems with the continuous versions, $E[R|R=r,x \in S_i]$ is equivalent to $E[R|R=r,x\in S_j]$, such that $r$ can be used as a variable representing a risk prediction value for a given group. However, we do correctly assume the dependence of $p^*$on $S$ in either case because we only use it in relation to $r$ and $\alpha$. In the updated manuscript, we have fixed this mistake and added a section showing that the discrete versions of MC,PMC,DC behave similarly to the continuous versions, and then worked with the continuous versions for simplicity.
>
> > It is unclear to me how the hyperparameters have been chosen for the experiments. This is however important to be able to evaluate the experiments correctly.
> >
>
> We agree with the importance and updated the text to make it more clear. It is for this reason we have evaluated the experiment under many different hyperparameters given in Table 2. These parameters were chosen to be generic - for example, $\alpha$ spans 3 orders of magnitude and $\rho$ spans two. a $\gamma$ of 0.05,0.1 were chosen from Barda et al, and seemed like reasonable lower probabilities for groups. similary, $\lambda$=0.1 was chosen for teh convention of 10 prediction bins. A larger $\lambda$ would simply be less expressive in postprocessing. For each parameter combination (64), we evaluated each method (base, MC, PMC) with 100 bootstrap random samples.
>
> > The main paper lacks a more thorough discussion/comparison of MC vs PMC in the experimental part. As it seems that MC with small values of can do as well as PMC for the various performance losses, I understand that in practice, the main advantage of PMC vs MC would be in terms of computational time budget. Do I understand correctly? It would be useful to discuss this in more depth in the main paper.
> >
>
> Yes, the reviewer is correct. We have moved much of the appendix text discussing this comparison into the Results section.
>
>
> > It is unclear to me what is plotted in Figure 1.
> >
>
> We have updated the labels to make it more clear. Figure 1 is a visualization of the relationships between the measures. the x-axis gives the value of a model that is, e.g., $\alpha$-multicalibrated. The y-axis gives the corresponding lower bound on that model’s other fairness measures, according to our theorems. We have moved this figure to the appendix in the revision to make room for other reviewer requests in the main text.
>
> > The main paper is not self-contained. The losses used to measure experimental performances are only provided in Appendix.
> >
>
> The sample loss functions take up about half a page, and it is extremely difficult to fit them. We have added text describing them in words. If there is something in the main text that seems less important we are open to revisiting this decision.
>
> > Tables (eg Table 3) are referred to in the main text without specifying that they are in Appendix.
> >
>
> Table 3 is indeed in the main text, page 9 of the original submission. There may be a cross link issue? We did not find one.
>
> > Theorem 4 states they there exists an algorithm for proportional multicalibration in a polynomial number of steps, then Algorithm 1 is presented as an algorithm which satisfies Theorem 4. It would sound more natural to me to introduce Algorithm 1, and then have a Proposition which states that it converges in a polynomial number of steps.
> >
>
> We have made the suggested change.
>
>
> Please let us know if there is anything else we can address that may improve your assessment of this work. Thanks again for your time and careful reading.

---

### Official Review · Reviewer_Csmw · 2022-10-26

**Confidence:** 5
**Clarity, Quality, Novelty And Reproducibility:** See above.
**Correctness:** 2
**Technical Novelty And Significance:** 2
**Empirical Novelty And Significance:** 3
**Recommendation:** 3

**Strength And Weaknesses:**

See above.

**Summary Of The Paper:**

Multicalibration requires calibration of a predictor even conditioned on membership in S where S comes form a family of protected subsets. Formally, we require $E[y] = E[f(x)]$ conditioned on x belonging to S and $f(x) \approx p$. If we relax this to allow $E[(y - f(x))] = \alpha$, conditioned on $f(x) \approx p$, we have $E[y] = p(1 \pm \alpha/p\mu(p))$ where $\mu(p)$ is the probability that $f(x) \approx p|x \in S$.

This paper posits that to achieve the fairness notion of sufficiency, one cares that we have a good multiplicative guarantee across all protected sets. Doing so requires
1. The set $S$ to be reasonably large, so that $\Pr(x \in S) \geq \gamma$.
2. The event $f(x) \approx p|x \in S$ to be reasonably large, in other words that $\mu(p) \geq \delta$.
3. The value $p$ itself is not too small.

Under these  conditions, multicalibration guarantees that condition of $S$, for most values $p$ (all but $\alpha$ fraction), the expected value of $y$ is multiplicatively close to our predictions.

This paper proposes a new notion called proportional MC that achieves this guarantee. But it seems that it can  be achieved by setting parameters differently in the original definition, so I am not convinced that this is really a new notion (or that one is needed).  Indeed, if you look at Definition 1 and 3, you can infer 3 from 1 by taking $\alpha = f(\alpha, \lambda,\rho)$. In other words, sufficiently small error in the standard notion ensures the multiplicative notion that is desired here.

At a higher level, having only sample access means that the best guarantees one could hope for are of the form $$ E[I(x, f(x))(y -f(x))] \leq \alpha$$
where $I$ is some event that depends on $x, f(x)$. Here it is $x \in S, f(x) \approx p$. If we wish to get multiplicative closeness under conditioning, we need:
1. A lower bound $\mu $ on how large E[I(x, f(x))] can be.
2. Take the multi calibration error $\alpha$ small enough so that $\alpha/\mu$ is meaningful.

This suffices the ensure that $E[y|I]$ and $[f(x)|I]$ are close. Here, since they want to error to be comparable to $p$, we also lower bound $p$.









**Summary Of The Review:**

The message that this paper makes that if one wants strong sufficiency guarantees, one has to be mindful of possibly low probability events is a good one. But I don't think this calls for a new definition, rather just a judicious setting of parameters in the existing one.

---

> ### Author Response · Authors · 2022-11-14
> **we demonstrate proportional mutlicalibration is a more effective way to achieve multiplicative loss**
>
> We thank the reviewer for taking the time to review our paper.
>
> > The message that this paper makes that if one wants strong sufficiency guarantees, one has to be mindful of possibly low probability events is a good one. But I don't think this calls for a new definition, rather just a judicious setting of parameters in the existing one.
>
> We are glad this interested the reviewer, and we hope to build a convincing case that our approach is a significantly better way to handle multiplicative closeness than the existing multicalibration definition.
>
> > Correctness: 2: Several of the paper’s claims are incorrect or not well-supported.
> >
>
> The reviewer has gave us a very low correctness score, but the review didn't contain any examples of incorrect claims made in the paper. We would appreciate the opportunity to address any mistakes.
>
> > This paper proposes a new notion called proportional MC that achieves this guarantee. But it seems that it can be achieved by setting parameters differently in the original definition, so I am not convinced that this is really a new notion (or that one is needed). Indeed, if you look at Definition 1 and 3, you can infer 3 from 1 by taking $\alpha = f(\alpha, \gamma, \rho)$. In other words, sufficiently small error in the standard notion ensures the multiplicative notion that is desired here.
> >
>
> > The message that this paper makes that if one wants strong sufficiency guarantees, one has to be mindful of possibly low probability events is a good one. But I don't think this calls for a new definition, rather just a judicious setting of parameters in the existing one.
> >
>
> We hope to convince the reviewer of the utility of PMC based on this exact observation. The reviewer correctly points out that multiplicative closeness might be achieved by setting $\alpha$ to be extremely low. It is for this reason that we looked at 3 orders of magnitudes of $\alpha$, included the best performance of any setting in Table 3 of the original paper. Furthermore, we expanded on this topic in the appendix (see page 20, and figures 5-7, and table 5 in the original submission). Quoting that section:
>
> > The ability of MC-postprocessing to perform well in terms of PMC and DC loss for certain
> values of $\alpha$  makes intuitive sense. If can be made small enough, the calibration error on all categories will be small compared to the outcome probability, $E[y|R \in I, x \in S]$ . However, to achieve this performance by MC postprocessing may require a large number of unnecessary updates for high risk intervals, since the DC and PMC of multicalibrated models are limited by low-risk groups (Theorem 1). Furthermore, the number of steps in MC-postprocessing (and PMC-postprocessing) scales as an inverse high-order polynomial of $\alpha$ (cf. Thm. 2 (Hebert-Johnson et al., 2018)).
> >
>
> In figures 5-7 and Table 5 of the original submission, we find that PMC post-processing achieves its lowest multiplicative loss, across values of $\alpha$,  in significantly fewer steps than MC post-processing. The lowest multiplicative loss achieved by PMC is also lower than by MC, even with very small values of $\alpha$.
>
> In our revision, we have emphasized these results more heavily in the main text (cf Table 3,4, Results section "sensitivity analysis"). Hopefully this demonstrates why PMC postprocessing is valuable for achieving multiplicative loss control versus MC postprocessing.
>
> ## guarantees given sample only access
>
> > If we wish to get multiplicative closeness under conditioning, we need:
> >
> > 1. A lower bound on how large E[I(x, f(x))] can be.
> > 2. Take the multi calibration error small enough so that is meaningful.
> >
> > This suffices the ensure that $E[y|I]$ and $[f(x)|I]$ are close. Here, since they want to error to be comparable to $p$, we also lower bound $p$.
> >
>
> We thank the reviewer for sharing these observations. However, we aren’t sure if this is just a comment, or a question or request related to the paper. Is the reviewer suggesting we provide an out-of-sample generalization analysis for PMC, as a function of $\alpha, \gamma, \rho$?

---

### Official Review · Reviewer_oBPB · 2022-10-30

**Confidence:** 3
**Correctness:** 3
**Technical Novelty And Significance:** 3
**Empirical Novelty And Significance:** 3
**Recommendation:** 6

**Clarity, Quality, Novelty And Reproducibility:**

Clarity: The paper is very well-written and easy to read. Some minor typos:

1. Page 4: That its-> That is. 1st sentence of 2.4: "MC constraints".

Novelty: The key claims made in the paper appear to be novel to me. The paper uses a lot of algorithmic machinery from previous papers on MC, but I believe that is not the key point of this paper.

Reproducibility: Detailed code and instructions to run the experiments are provided.

**Strength And Weaknesses:**

Strengths:

1. The authors point out a potentially important issue with calibration scores (the fact that they could lead to larger errors for groups with lower probability estimates). PMC seems like a good way to correct for this,

2. The authors establish an interesting connection between MC and notions of sufficiency and the new notion of differential calibration. The notion of sufficiency seems quite related to the concept of "Outcome indistinguishability" (https://arxiv.org/abs/2011.13426) and it would be nice if the authors elaborate a bit more on this connection.

3. I quite liked the fact that the authors evaluated their results on real-world data which comes close to capturing the problem the paper aims to solve.

Weaknesses:

1. The paper has a number of strengths, but I'm not fully convinced of the papers claim about the superiority of PMC to MC. To me, the choice of relative error vs additive errors seems to be closer to a normative judgement, or at the very least the two need to be compared with respect to how they affect decision making in the use cases. Can I not make the claim that the proposed relative error notion is unfair, since it allows for a larger (additive) error for groups which do have higher probabilities? Could a decision maker not end up having less trust in the model's predictions if it has higher errors for those groups? Additive error almost seems like a simpler to understand notion (since the error remains the same for all predicted probabilities), and I could be convinced to believe that it could lead to better downstream decision making because of this property. Perhaps the authors can point out contexts where calibration itself is more meaningfully defined as a relative notion rather than an additive notion? If these settings exists, then PMC naturally comes out as a better solution than MC.

2. I think it would be good if the authors commented a bit more on the fact that PMC seems to have lower accuracy than MC in the real-world experiment (I appreciate that the authors include this result). Is PMC trading-off increasing fairness guarantees for accuracy?

**Summary Of The Paper:**

The paper proposes an interesting new notion of "Proportional multicalibration" for algorithmic fairness. Proportional multicalibration (PMC) builds on the notion of multicalibration (MC). MC requires that the predictor is calibrated for each (identifiable) group within the data, as opposed to simply requiring calibration over the entire population. The notion of MC has been well-studied at this point, and has been found to be useful in practical contexts as well.

The authors point out a potential flaw in MC. MC is defined to allow an additive calibration error (following on the standard notion of calibration, which is also defined additively). The authors claim that this could be problematic since it allows for a larger relative calibration error for groups which have smaller probability estimates. PMC aims to correct for this, by requiring that the relative calibration error be bounded for all the groups. PMC also satisfies a notion of differential-calibration with better guarantees than MC.

The authors substantiate their theoretical results with an interesting experiment on real-world hospital data.

**Summary Of The Review:**

In summary, I quite liked reading this paper and I think it has interesting ideas. Even at the current stage, it would be an interesting addition to the ICLR program. Perhaps the authors could comment a bit more on the weaknesses I wrote earlier?

---

> ### Author Response · Authors · 2022-11-14
> **Thanks and response to weaknesses**
>
> We thank the reviewer for their time and for an encouraging review. If there are other points we can address that would help improve the assessment of our paper, we are available to make further changes and discuss.
>
> > *The paper has a number of strengths, but I'm not fully convinced of the papers claim about the superiority of PMC to MC.*
>
> Regarding the objective superiority of PMC over MC: we would not claim that one is superior to the other as an objective measure of error. Both are quite valuable measures, and it is for this reason we report both in our experiments.
> However, we would claim there are advantages to minimizing differentials/ratios of calibration error, including a) removing the dependence of MC on the outcome probability, which may correlate with group membership; b) aligning PMC with differential fairness measures and sufficiency, which levels the trustworthiness of the model across groups.
>
> > *Perhaps the authors can point out contexts where calibration itself is more meaningfully defined as a relative notion rather than an additive notion? If these settings exists, then PMC naturally comes out as a better solution than MC.*
> >
>
> The context we would point to is that in our simulation: the stronger that the outcome probability depends on group membership, the more important it is to fairness for a multiplicative notion of calibration equity to be satisfied. We have overhauled some sections of the paper to hopefully bring this across more clearly.
>
> > *Additive error almost seems like a simpler to understand notion (since the error remains the same for all predicted probabilities), and I could be convinced to believe that it could lead to better downstream decision making because of this property*
>
> Regarding interpretability: we believe there is an argument to make for PMC to be considered more interpretable. Percentage error is a standard way of talking about error that works across measures and scales. In contrast, additive errors cannot be thought of this way. So, we would argue PMC is potentially easier to understand than MC for many. E.g., a PMC-style guarantee like “this model is miscalibrated by at most 10%” may be quite understandable. That said, an MC-style guarantee like “this model is miscalibrated by at most 0.1” may also be easy to understand, depending on the recipient’s comfort with probabilities.
>
> > *Perhaps the authors can point out contexts where calibration itself is more meaningfully defined as a relative notion rather than an additive notion?*
>
> Calibration is often measured by the Brier score, which is equivalent to MSE($y$,$R$). The closest analogue for PMC that comes to mind is using mean absolute percentage error ([MAPE](https://en.wikipedia.org/wiki/Mean_absolute_percentage_error)) in place of the Brier score. The formulations differ from MC and PMC (we’re looking at differences in expectations versus expectation of differences), but the principal motivation is the same. MAPE is more commonly used in regression, often because it has a more intuitive interpretatation.
>
> > *I think it would be good if the authors commented a bit more on the fact that PMC seems to have lower accuracy than MC in the real-world experiment (I appreciate that the authors include this result). Is PMC trading-off increasing fairness guarantees for accuracy?*
> >
>
> For the RF models, PMC exhibits lower accuracy than MC. For the LR models, PMC exhibits higher accuracy than MC. In both cases, the overall difference is less than a tenth of a percent, so we feel the correct conclusion is that the effect of MC or PMC on AUROC is negligible, both for both RF and LR models.

---

> > ### Comment · Reviewer_oBPB · 2022-11-20
> > **Thank you for the response**
> >
> > Thanks to the authors for the detailed response.
> >
> > I believe the choice between an additive vs a multiplicative calibration error guarantee is probably domain dependent, but I can imagine that a multiplicative error guarantee as suggested here could possibly be more meaningful in certain scenarios. It would be good for the paper if the authors could outline a concrete scenario where this is the case (a scenario where having a larger multiplicative calibration error for groups with lower outcome probabilities is unfair).
> >
> > Overall, the paper contains an interesting idea and appears to be above the acceptance threshold for me.

---

### Author Response · Authors · 2022-11-16
**Summary of author response to initial reviews**

We want to thank the reviewers again for carefully reading our manuscript. The reviews have substantially improved our paper.

We would like to summarize what has been discussed and changed so far:

- For the most part, the reviewers found the paper interesting and well motivated. Over the remaining discussion period, we hope to convince all that it would be a valuable contribution to ICLR.

- Two reviewers correctly pointed out an issue with the proofs to Theorem 1 and 2 that has been fixed during this rebuttal period. The issue stemmed from the binning of predictions, which introduces uncertainty as to the expected risk score within any one bin and group. We fixed this issue by formulating the theorems and proofs using continuous predictions, as has been done in most prior work on multicalibration. Working with the continuous version of PMC simplifies the proofs but also assumes that, by minimizing the discrete version of PMC, one can achieve the continuous version. We have added additional analysis to show this is the case in Appendix A.2.1. In short, $(\alpha, \lambda)$-PMC equates to $(\alpha + \frac{\lambda}{\rho})$-PMC using fixed width bins, and equates to $(\alpha \rho^{-\lambda} + \rho^{-\lambda}-1)$-PMC using fixed-width bins on a log scale. This is the proportional analog to $(\alpha,\lambda)$-MC, which implies $(\alpha+\lambda)$-MC.

- Reviewer RcCa noted the importance of the issue of satisfying sufficiency among groups with low outcome probability, but wasn't convinced a new definition of MC was necessary when one could set $\alpha$ in the original definition very small to achieve multiplicative closeness. In response, we moved our experiments and discussion of this point to the main text from the appendix. In short, MC can achieve multiplicative closeness for very small values of $\alpha$, but it requires a very large number of updates. The large number of updates arise from not distinguishing between low- and high-probability outcomes among groups - which is directly addressed by using the PMC definition and algorithm in its place. In practice this reduces the number of updates by about 5x (Table 4). Aside from the number of evals, the best performance according to multicalibration error and proportional multicalibration error is almost always achieved by PMC-postprocessing, using the best set of hyperparameters for each algorithm (Table 3).

- We have made several edits to improve clarity and add detail to the expeirments. We hope to have answered each of the reviewer's comments, questions and criticisms.

- If there are remaining / outstanding concerns, we remain available to discuss them in the final few days of this discussion period.

---

### Decision · Program_Chairs · 2023-01-20

**Decision:**

Reject

**Justification For Why Not Higher Score:**

The novelty appears to be a significant issue.

**Justification For Why Not Lower Score:**

N/A

**Metareview: Summary, Strengths And Weaknesses:**

This work develops a new notion of "Proportional Multicalibration" (PMC) for algorithmic fairness, building on the well-studied notion of multicalibration (MC). MC requires that the predictor is calibrated for each (identifiable) group within the data, as opposed to simply requiring calibration over the entire population. MC is defined to allow an additive calibration error: the claim in this paper is that this could be problematic since it allows for a larger relative calibration error for groups that have smaller probability estimates. PMC aims to correct for this, by requiring that the relative calibration error be bounded for all the groups: to achieve the fairness notion of sufficiency, the claim made is that we need a good multiplicative guarantee across all protected sets.

This paper basically claims that if we want strong sufficiency guarantees, we have to be mindful of possibly low-probability events. As a reviewer points out, this often calls for just a judicious setting of parameters in the existing definition: e.g., multiplicative closeness might be achieved by setting alpha very low.

The authors are encouraged to discuss the novelty of their work more clearly, and also present concrete settings where a multiplicative error guarantee would be much more meaningful.